# Functional Variational Inference based on Stochastic Process Generators

**Chao Ma**
University of Cambridge
Cambridge, UK
cm905@cam.ac.uk

**José Miguel Hernández-Lobato**
University of Cambridge
Cambridge, UK
jmh233@cam.ac.uk

## Abstract

Bayesian inference in the space of functions has been an important topic for Bayesian modeling in the past. In this paper, we propose a new solution to this problem called Functional Variational Inference (FVI). In FVI, we minimize a divergence in function space between the variational distribution and the posterior process. This is done by using as functional variational family a new class of flexible distributions called Stochastic Process Generators (SPGs), which are cleverly designed so that the functional ELBO can be estimated efficiently using analytic solutions and mini-batch sampling. FVI can be applied to stochastic process priors when random function samples from those priors are available. Our experiments show that FVI consistently outperforms weight-space and function space VI methods on several tasks, which validates the effectiveness of our approach.

## 1 Introduction

As an important approach to Bayesian deep learning, Bayesian neural networks (BNNs) have been proposed and studied for decades [29, 34, 19]. Despite some successful applications in specific cases [55, 44, 23], BNNs still suffer from several shortcomings. As over-parameterized models, BNNs often have multiple posterior modes in its weight space, that generate identical predictions [30]. Therefore, Bayesian inference in weight space can be very difficult [50]. Furthermore, due to the complexity of interactions between weights in the network forward pass, it is unclear the effect that a given prior distribution over the weights will have in the resulting distribution over functions.

To address these issues, there has been a recent resurgence of interest in applying the perspective of Bayesian non-parametrics to neural nets. That is, BNNs are treated as probability measures over *functions*, i.e., as stochastic processes, with Bayesian inference being performed now in *function space*.e Examples include functional BNNs (f-BNNs) [47], and variational implicit processes (VIPs) [28]. Despite their empirical advantage over weight space VI, they still suffer from a number of issues. i), f-BNNs optimize a functional KL divergence between BNNs and GPs, which is not always well-defined[5]. Also, f-BNNs rely on (spectral) stein gradient estimators, which are less efficient for high dimensional distributions [56, 14]. While VIPs do not have this problem, they use a wake-sleep procedure that does not correspond to a single unified objective. ii), F-BNNs rely on a mean-field approximation, which often lacks predictive in-between uncertainty on test data. This issue was observed both with single layer BNNs [10], and deeper BNNs (able to represent in-between uncertainty, but more over-confident than HMC empirically for certain settings [11, 9]). On the other hand, VIPs use as variational family a Gaussian process, which is not able to capture non-Gaussian processes such as structured implicit priors [47].

Therefore, it is an open challenge how to improve and justify (variational) inference in the space of functions using priors given by stochastic processes. In this paper, we investigate this old but important problem and propose a new solution. Our contributions are as follows:

35th Conference on Neural Information Processing Systems (NeurIPS 2021).

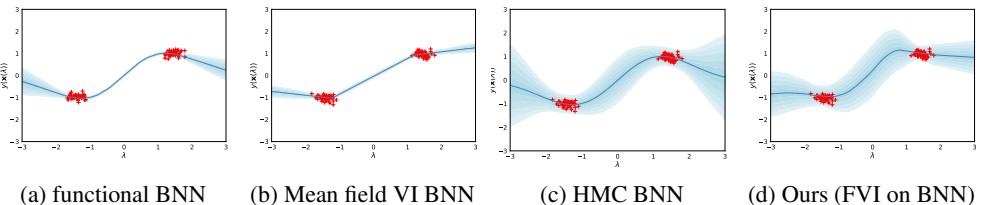

| (a) functional BNN | (b) Mean field VI BNN | (c) HMC BNN | (d) Ours (FVI on BNN) |

Figure 1: A regression task on a synthetic dataset (red crosses) from [10]. We plot predictive mean and uncertainties for each algorithm. This tasks is used to demonstrate the theoretical pathologies of weight-space VI for single-layer BNNs: there is *no* setting of the variational parameters that can model the in-between uncertainty between two data clusters. The functional BNNs [47] also have this problem, since mean-field BNNs are used as part of the model. On the contrary, our FVI method can produce sensible uncertainty estimates. See Appendix C.2 for more details.

- We propose a new objective function for variational inference in function space, as an alternative to functional KL divergence between stochastic prcoesses [47]. We show that this new objective is a valid divergence, and can avoid some of the problems that the functional KL divergence has.

- We propose a new class of flexible variational distributions in function space, called stochastic process generators (SPGs). SPGs are non-Gaussian generalizations of the VIP family [28], and can help avoid the fundamental limitation of mean field Gaussians [10] used in BNNs and functional BNNs. A theorem regarding the expressiveness of SPGs is proved (Proposition 4). Based on SPGs, our proposed functional divergence between stochastic processes can be estimated efficiently using mini-batch sampling (Proposition 5, 6 and 7), which achieves a significant speed-up against the gradient estimator approach in [47].

- We compare our methods against existing weight-space and function-space inference methods in several tasks. Our method consistently outperforms the baselines, and is much faster than f-BNN, which validates the effectiveness of our approach.

## 2 Backgrounds

**Variational BNNs**    Let $y = g(\mathbf{x}, \mathbf{w})$ be a neural network, where $\mathbf{x}$ is the input, $y$ is the output, and $\mathbf{w}$ denotes the weights. To build a Bayesian neural network (BNN), we place a prior $p(\mathbf{w})$ over $\mathbf{w}$, and add an observational noise $\epsilon \sim \mathcal{N}(\epsilon; 0, \sigma^2)$ to the output. Then, a BNN is given by

$$\log p(y|\mathbf{x}) = \log \int_{\mathbf{w}} \mathcal{N}(y; g(\mathbf{x}, \mathbf{w}), \sigma^2) p(\mathbf{w}) d\mathbf{w}. \tag{1}$$

Given a dataset $\mathcal{D} = \{\mathbf{x}_i, y_i\}_{i=1}^N$, the goal of Bayesian inference is to compute the posterior $p(\mathbf{w}|\mathcal{D}) \propto p(\mathcal{D}|\mathbf{w})p(\mathbf{w})$. Since this is intractable, we often resort to approximate inference methods such as variational inference (VI), e.g. Bayes by Backprop (BBB) [4], variational dropout [12], or other extensions [20, 26, 41]. Since the posterior $p(\mathbf{w}|\mathcal{D})$ is usually highly multi-modal due to the non-identifiability of neural networks, VI often leads to unsatisfactory results, and suffers from over-parameterization and other pathologies [10].

**Variational Implicit Processes (VIPs)**    An alternative way of looking at BNNs is to treat them as a prior distribution over *function space*, i.e., a *stochastic process*. Indeed, $p(y|\mathbf{x})$ in Equation 1 generates a random function $g(\cdot, \mathbf{w})$, depending on what $\mathbf{w}$ is actually sampled from the prior. This idea is further explored by the variational implicit process (VIPs). Given a prior $p(f)$ on random functions, VIP approximates the posterior in function space $p(f|\mathcal{D})$ by using a Bayesian linear regression model $q_{\text{VIP}}(f|\mathcal{D}) = \sum_s a_s \phi_s, \mathbf{a} \sim \mathcal{N}(\mathbf{a}; \mu, \mathbf{\Sigma})$, where the basis functions $\{\phi_s\}_{s=1}^S$ are random samples drawn directly from $p(f)$. Although VIP has shown improved performance over parameter-space VI methods, it has a few limitations. For example, $q_{\text{VIP}}(f|\mathcal{D})$ resembles a GP approximation to $p(f|\mathcal{D})$, whereas the true posterior in function space might be arbitrarily complex. Also, the wake-sleep training procedure of VIP does not optimize any coherent objective function. Lastly, VIP requires the prior $p(f)$ to be reparameterizable. i.e., can be represented in the form of $f(\mathbf{x}) = g_\theta(\mathbf{x}, \mathbf{z}), \mathbf{z} \sim p(\mathbf{z})$. This may limit its application to non-reparameterizable priors.

## 3   Problem setting and the functional KL divergence

We consider the problem of Bayesian inference in function space. Let $p(f)$ be a stochastic process defined on the probability space $(\Omega, \mathcal{B})$. Note that we use $f$ in its scalar form to denote a scalar function $f(\cdot) : \mathcal{T} \mapsto \mathbb{R}$. Here $\mathcal{T}$ is the index set of $p(f)$ (assumed to be a compact subset of $\mathbb{R}^d$). For example, $p(f)$ could be a Gaussian process $p(f) = \mathcal{GP}(m(\cdot), k(\cdot, \cdot))$, a Bayesian neural network, or any other suitable stochastic process. We use $p(f)$ to model the uncertainty in function space. Then, a likelihood function $p_\pi(y|f(\cdot))$ is defined on top of $f$ to generate observable data $y$.

Given observed data $\mathcal{D} = \{\mathbf{x}_i, y_i\}_{i=1}^N$, our goal is to infer the posterior process $p(f|\mathcal{D})$ conditioned on the observations $\mathcal{D}$. If $p(f)$ is a GP, then $p(f|\mathcal{D})$ can be computed analytically. However, in most cases this is intractable. Therefore, following [32, 47], we define another stochastic process $q(f)$ on $(\Omega, \mathcal{B})$ as our variational family to approximate $p(f|\mathcal{D})$. $q(f)$ can be optimized by maximizing the evidence lower bound (ELBO) in *function space*:

$$\mathcal{L}_q^{functional} := \mathbb{E}_{q(f)}[\log p_\pi(\mathcal{D}|f)] - D_{KL}[q(f)||p(f)]. \tag{2}$$

Note that $D_{KL}[q(f)||p(f)]$ is the *functional KL-divergence between stochastic processes*. As shown by [47], $D_{KL}[q(f)||p(f)]$ can be written as:

$$D_{KL}[q(f)||p(f)] = \sup_{n,\mathbf{X}_n} D_{KL}[q(\mathbf{f}^{\mathbf{X}_n})||p(\mathbf{f}^{\mathbf{X}_n})], \tag{3}$$

where $\mathbf{X}_n$ denote a set of $n$ measure points $\{\mathbf{x}_k\}_{1 \le k \le n}$ in the domain/index set of $f(\cdot)$, which can be treated as an element of the product space $\mathcal{T}^n$. Moreover, $\mathbf{f}^{\mathbf{X}_n}$ denotes the vector of function values evaluated on $\mathbf{X}_n$, and $D_{KL}[q(\mathbf{f}^{\mathbf{X}_n})||p(\mathbf{f}^{\mathbf{X}_n})]$ is the KL-divergence over random vectors typically used by the machine learning community. In other words, the KL-divergence between stochastic processes is the supreme of the relative entropies obtained on all possible measure points in $\mathcal{T}^{\mathbb{Z}^+}$.

## 4   Functional Variational Inference using Stochastic Process Generators

### 4.1   The grid-functional KL divergence

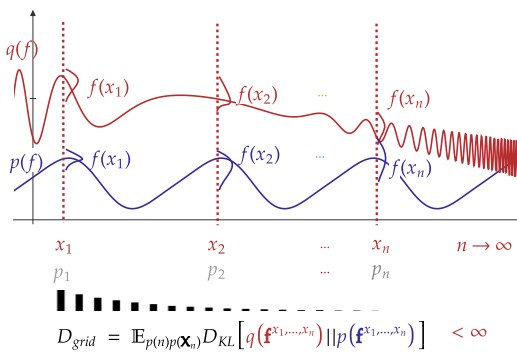

Figure 2: Illustration of $\mathcal{D}_{grid}$

As pointed out by [5], the functional KL divergence defined in Eq 3 is not always well-defined. For example, the functional KL divergence between two BNNs with different network architectures can be infinite. To address this issue, we propose to optimize a new divergence measure called the *grid-functional KL divergence*. Figure 2 illustrates the idea of grid functional KL divergence, which we now proceed to describe. Instead of taking the supremum as in the original functional KL divergence, we take an expectation over *both* $n$ and $\mathbf{X}_n$. We specify a probability distribution $\{p_n\}_{1 \le n < \infty}$ over $n$ assigning lower probability values $p_n$ to larger $n$. This way, we can trim down the contributions from the KL terms $D_{KL}[q(\mathbf{f}^{\mathbf{X}_n})||p(\mathbf{f}^{\mathbf{X}_n}|\mathcal{D})]$ that correspond to large $n$, and hopefully arrive at a finite expectation value. Following this idea, we give the formal definition of grid-functional KL divergence as

$$\mathcal{D}_{grid}[q(f)||p(f|\mathcal{D})] := \mathbb{E}_{n,\mathbf{X}_n \sim c} D_{KL}[q(\mathbf{f}^{\mathbf{X}_n})||p(\mathbf{f}^{\mathbf{X}_n}|\mathcal{D})], \tag{4}$$

where $\mathbf{X}_n$ is a set of $n$ measurement points $\{\mathbf{x}_k\}_{1 \le k \le n}$ sampled from $\mathcal{T}$, according to some sampling distribution $c$. Note that $c$ is defined on the product space $\mathcal{T}^{\mathbb{Z}^+}$ and the number of sampled measure points $n$ is also random. One may recognize that [47] proposed a similar objective as an approximation to 3 in which the number of measure points $n$ is a fixed constant instead of a random variable. However, as we show below, it is in fact critical that $n$ is *not* fixed.

**Proposition 1.** *Suppose $c$ has full support on $\mathcal{T}^{\mathbb{Z}^+}$. Then, $\mathcal{D}_{grid}[q(f)||p(f|\mathcal{D})]$ satisfies the following conditions: i), $\mathcal{D}_{grid}[q(f)||p(f|\mathcal{D})] \ge 0$; ii), $\mathcal{D}_{grid}[q(f)||p(f|\mathcal{D})] = 0$ if and only if $q(f) = p(f|\mathcal{D})$.*

In other words, if $c$ has full support on $\mathcal{T}^{\mathbb{Z}^+}$ (we will give an example of such $c$ later), then $\mathcal{D}_{grid}[q(f)||p(f|\mathcal{D})]$ is a valid divergence in function space. Therefore, we can use $\mathcal{D}_{grid}$ as an alternative objective for function space inference. We can show that, for certain scenarios, $\mathcal{D}_{grid}$ can avoid some of the issues the original functional KL divergence has (see Appendix A.2 for details):

**Proposition 2.** *Let $p(f)$ and $q(f)$ be two distributions for random functions with different parametric forms. Assume that $p(f)$ is parameterized by the following sampling processes:*

$$f = g + \epsilon, g(\mathbf{x}) \sim p(g|\mathbf{x}; \Theta), \Theta \sim p(\Theta), \epsilon \sim \mathcal{N}(0, \sigma^2)$$

*and $q(f)$ is parameterized by:*

$$f = g + \epsilon, g(\mathbf{x}) \sim q(g|\mathbf{x}; \Gamma), \Gamma \sim q(\Gamma), \epsilon \sim \mathcal{N}(0, \sigma^2).$$

*Here, $\mathbf{x} \in \mathcal{T} \subset \mathbb{R}^d$ is the input variable of the $f$, $g$ is the random latent function, and $\Theta \in \mathbb{R}^I$, $\Gamma \in \mathbb{R}^J$ are the parameters of each random function, respectively. Suppose $p(g|\mathbf{x}; \Theta)$, $q(g|\mathbf{x}; \Theta)$, $p(\Theta)$ and $q(\Gamma)$ all have compact supports w.r.t. $g$, $g$, $\Theta$, and $\Gamma$, respectively. Then, there exist a sampling distribution $c$ such that: 1) $c$ has full support on $\mathcal{T}^{\mathbb{Z}^+}$, and 2) $\mathcal{D}_{grid}[q(f)||p(f)]$ is finite.*

This result shows that the grid-functional KL divergence allows us to perform VI between $p$ and $q$ even if they have *different* parametric forms. In Appendix A.2, we further show that the compactness assumptions can be relaxed under certain assumptions. Moreover, in Appendix A.3 Corollary 1, we have shown that if one of the distributions is replaced by a Gaussian process (of certain kernel function), then under some additional assumptions, the grid-functional KL is still finite. In those cases, the original functional KL is no longer finite. This validates our choice of grid-functional KL divergence. To use $\mathcal{D}_{grid}[q(f)||p(f|\mathcal{D})]$ for VI, we further derive a new ELBO based on $\mathcal{D}_{grid}$ (Appendix A.1):

**Proposition 3.** *Let $n, \mathbf{X}_n \sim c$ be a set of random measure points such that $\mathbf{X}_\mathcal{D} \subset \mathbf{X}_n$. Define*

$$\mathcal{L}_q^{grid} := \log p(\mathcal{D}) - \mathcal{D}_{grid}[q(f)||p(f|\mathcal{D})]. \tag{5}$$

*Then we have*

$$\mathcal{L}_q^{grid} = \mathbb{E}_{q(f)}[\log p(\mathcal{D}|f)] - \mathcal{D}_{grid}[q(f)||p(f)] \tag{6}$$

*and $\log p(\mathcal{D}) \geq \mathcal{L}_q^{grid} \geq \mathcal{L}_q^{functional}$.*

Proposition 3 shows that $\mathcal{L}_q^{grid}$ is a valid variational objective function: it is a lower bound for $\log p(\mathcal{D})$ and also upper-bounds $\mathcal{L}_q^{functional}$. For the rest of the paper, we will discuss how to perform functional VI based on $\mathcal{L}_q^{grid}$. We will focus on how to propose a expressive variational family $q(f)$ and how to efficiently estimate $D_{grid}[q(f)||p(f)]$.

**Remark.** One example of $c$ that satisfies the requirement of Propositions 1, 2, and 3 takes the following form (which will be used throughout the paper):

$$(n - |\mathcal{D}|) \sim \text{Geom}(p), \mathbf{x}_k \sim \mathcal{U}(\mathcal{T}), \ \forall 1 \leq k \leq n - |\mathcal{D}|, \mathbf{X}_n := \mathbf{X}_\mathcal{D} \bigcup \{\mathbf{x}_k\}_{1 \leq k \leq n - |\mathcal{D}|}, \tag{7}$$

where we first sample $n$ from a geometric distribution such that $(n - |\mathcal{D}|) \sim \text{Geom}(p)$ with parameter $p$ (see Appendix A.2 for more discussion). Then, $(n - |\mathcal{D}|)$ out of distribution (OOD) measure points are sampled independently from a uniform distribution on $\mathcal{T}$.

### 4.2 Choosing $q(f)$: stochastic process generators

In order to obtain good performance in FVI, it is crucial to pick an expressive variational family for $q(f)$. Here, we propose a new class of variational distributions called stochastic process generator (SPG). SPGs can be seen as the non-Gaussian extension of the variational approximation used in VIPs. Recall that in VIPs, the variational family $q_{\text{VIP}}(f)$ is defined by the following sampling process:

$$f = \sum_s a_s \phi_s(\cdot), \quad \mathbf{a} \sim \mathcal{N}(\mathbf{a}; \mu, \boldsymbol{\Sigma}), \tag{8}$$

where $\{\phi_s\}_{s=1}^S$ are random paths sampled from the prior process $p(f)$. In SPGs, we remove the Gaussian assumption on $\mathbf{a}$. This is done by specifying a separate variational auto-encoder to represent the non-Gaussian distribution for $\mathbf{a}$:

$$q(\mathbf{a}) = \int_\mathbf{h} p_\theta(\mathbf{a}|\mathbf{h}) q_\eta(\mathbf{h}), \tag{9}$$

where $\mathbf{h}$ is the latent variable of the VAE, $p_\theta(\mathbf{a}|\mathbf{h})$ is the decoder parameterized by $\theta$, and $q_\eta(\mathbf{h})$ is a distribution over the latent space. Furthermore, we remove the constraint that $\{\phi_s\}_{s=1}^S$ need to be sampled from $p(f)$. For example, we may assume that each $\phi_s : \mathbb{R}^d \mapsto \mathbb{R}^1$ is a DNN with learnable weights $\mathbf{w}_s$. We can finally give the definition of our SPG variational family $q_{\text{SPG}}(f|q_\eta(\mathbf{h}))$:

$$f = \sum_s a_s \phi_s(\cdot, \mathbf{w}_s) + \nu, \quad \mathbf{a} \sim \int_{\mathbf{h}} p_\theta(\mathbf{a}|\mathbf{h}) q_\eta(\mathbf{h}), \quad \nu \sim \mathcal{GP}(\nu; 0, \delta(\cdot, \cdot)\sigma_\nu^2), \qquad (10)$$

where $\nu$ is a white noise process that models additive aleatoric uncertainty. Intuitively, $q_{\text{SPG}}$ serves as a generator for stochastic processes. $q_{\text{SPG}}(f|\cdot)$ maps any given $q_\eta(\mathbf{h})$ to a stochastic process $q_{\text{SPG}}(f|q_\eta(\mathbf{h}))$, hence the name. Regarding the expressiveness of SPGs, we have the following result:

**Proposition 4** (Expressiveness of SPGs). *Let $p(f)$ be a square-integrable stochastic process on a probability space $(\mathcal{X}, \mathcal{B})$, and its index set $\mathcal{T}$ is a compact subset of $\mathbb{R}^d$. Suppose $\mathcal{X}$ is a compact metric space, $\mathcal{B}$ is the Borel set on $\mathcal{X}$. Then, for $\forall \epsilon > 0$, there exists a SPG $q_{SPG}^\epsilon(f)$, such that:*

$$\text{MMD}(p, q_{SPG}^\epsilon; \mathcal{F}) < \epsilon \quad \text{for} \quad \forall \mathbf{x} \in \mathcal{T}, \qquad (11)$$

*where $\text{MMD}$ is the maximum mean discrepancy, $\mathcal{F}$ is the MMD function class defined to be a unit ball in a RKHS with a universal kernel [46] $k(\cdot, \cdot)$ as its reproducing kernel.*

Next we will discuss how $\{\mathbf{w}_s\}$, $\theta$ and $q_\eta(\mathbf{h})$ can be estimated and how can we use SPGs to estimate the function space KL-divergence in Equation 3.

### 4.3 Efficient estimation of grid-functional KL

In order to estimate the grid-functional KL-divergence in Equation 4, we propose to use a two-step method for a scalable approximation. In the first step, we distill $p(f)$ by fitting a stochastic process generator $\tilde{p}_{\text{SPG}}(f)$ to $p(f)$. In the second step, we calculate the KL-divergence between $q_{\text{SPG}}(f)$ and $\tilde{p}_{\text{SPG}}(f)$ as the surrogate for the KL-divergence between $q_{\text{SPG}}(f)$ and $p(f)$.

**Distilling $p(f)$ via a stochastic process generator**. Assume that we can draw $M$ random functions $[f_1(\cdot), f_2(\cdot), ..., f_m(\cdot), ..., f_M(\cdot)]$ from the prior process $p(f)$. Let $\tilde{p}_{\text{SPG}}(f)$ be an SPG given by

$$f = \sum_s a_s \phi_s(\cdot, \mathbf{w}_s) + \nu, \quad \mathbf{a} \sim \int_{\mathbf{h}} p_\theta(\mathbf{a}|\mathbf{h}) p_0(\mathbf{h}), \quad \nu \sim \mathcal{GP}(\nu; 0, \delta(\cdot, \cdot)\sigma_\nu^2), \qquad (12)$$

where $p_0(\mathbf{h})$ is a fixed standard normal distribution. We denote the above process by $\tilde{p}_{\text{SPG}}(f|p_0(\mathbf{h}))$. We can train $\tilde{p}_{\text{SPG}}(f)$ on $f_1, f_2, ..., f_m, ..., f_M$, by optimizing the aggregated ELBO on $\tilde{p}_{\text{SPG}}(f)$:

$$\max_{\{\mathbf{w}_s\}, \theta, \lambda} \mathbb{E}_{\mathbf{X}_O} \sum_m \log \tilde{p}_{\text{SPG}}(\mathbf{f}_m^{\mathbf{X}_O}) \geq \max_{\{\mathbf{w}_s\}, \theta, \lambda} \mathbb{E}_{\mathbf{X}_O} \sum_m \mathbb{E}_{\tilde{q}_\lambda(\mathbf{h}|\mathbf{f}_m^{\mathbf{X}_O})} \log \frac{\tilde{p}_{\text{SPG}}(\mathbf{f}_m^{\mathbf{X}_O}|\mathbf{h}) p_0(\mathbf{h})}{\tilde{q}_\lambda(\mathbf{h}|\mathbf{f}_m^{\mathbf{X}_O})}, \qquad (13)$$

where $\mathbf{f}_m^{\mathbf{X}}$ are the function values of $f_m$ evaluated on $\mathbf{X}_O$, $\mathbf{X}_O$ are $|O| \leq |\mathcal{D}|$ measure points independently sampled from the training set $\mathbf{X}_\mathcal{D} = \{\mathbf{x}_i\}_{i=1}^N$. $\mathbf{f}_m^{\mathbf{X}}$ are the function values of $f_m$ evaluated on $\mathbf{X}_O$, and $\tilde{q}_\lambda(\mathbf{h}|\mathbf{f}_m^{\mathbf{X}_O})$ is an encoder network that approximates the true posterior $\tilde{p}_{\text{SPG}}(\mathbf{h}|\mathbf{f}_m^{\mathbf{X}_O})$.

**Product of Experts (PoE) encoder**. When sampling $\mathbf{X}_O$, since its size might vary each time, we would need to set up to $2^N$ inference nets, one for each possible subsets. To overcome this issue, we adopt the Product of Experts encoder [53], a simple and flexible approach for such scenario given by

$$\tilde{q}_\lambda(\mathbf{h}|\mathbf{f}_m^{\mathbf{X}_O}) \propto p_0(\mathbf{h}) \prod_{i=1}^{|O|} \tilde{q}_\lambda(\mathbf{h}|f_m(\mathbf{x}_i), \mathbf{x}_i), \qquad (14)$$

where $\tilde{q}_\lambda(\mathbf{h}|f_m(\mathbf{x}_i), \mathbf{x}_i)$ is an inference network representing the expert associated with the $i$-th measurement point. Now the we can use a single encoder $\tilde{q}_\lambda(\mathbf{h}|\mathbf{f}_m^{\mathbf{X}_O})$ to handle all the possible inputs $\mathbf{f}_m^{\mathbf{X}_O}$. In practice, we let $\tilde{q}_\lambda(\mathbf{h}|f_m(\mathbf{x}_i), \mathbf{x}_i)$ to be a Gaussian expert that maps $[f_m(\mathbf{x}_i), \mathbf{x}_i]$ to a factorized Gaussian in latent space. Since the product of Gaussian experts is still Gaussian, $\tilde{q}_\lambda(\mathbf{h}|\mathbf{f}_m^{\mathbf{X}_O})$ is a Gaussian distribution whose statistics can be computed analytically.

**Estimating the grid-functional KL-divergence given $\mathbf{X}_n$**. In order to estimate the grid-functional KL divergence between $q_{\text{SPG}}(f)$ and $p(f)$, we first discuss how this divergence can be estimated on *measurement points* $\mathbf{X}_n$, i.e., $D_{KL}[q(\mathbf{f}^{\mathbf{X}_n})||p(\mathbf{f}^{\mathbf{X}_n})]$ where $\mathbf{f}^{\mathbf{X}_n}$ is the vector of function values

evaluated on $\mathbf{X}_n$. We then discuss how this can be used to estimate the grid-functional divergence in Equation 4. To begin with, as in Section 4.2, our variational family is given by

$$f = \sum_s a_s \phi_s(\cdot, \mathbf{w}_s) + \nu, \quad \mathbf{a} \sim q(\mathbf{a}) = \int_{\mathbf{h}} p_\theta(\mathbf{a}|\mathbf{h}) q_\eta(\mathbf{h}), \quad \nu \sim \mathcal{GP}(\nu; 0, \sigma_\nu^2). \tag{15}$$

We denote the above variational family by $q_{\text{SPG}}(f|q_\eta(\mathbf{h}))$. The key ingredient of our estimation method is that we force $q_{\text{SPG}}(f|q_\eta(\mathbf{h}))$ and $\tilde{p}_{\text{SPG}}(f|p_0(\mathbf{h}))$ to share the same basis functions (or weights $\{\mathbf{w}_s\}$) and decoder parameters $\theta$. That is, once optimal $\{\mathbf{w}_s\}$ and $\theta$ are obtained by fitting $\tilde{p}_{\text{SPG}}(f)$ to $p(f)$, these are frozen and reused in $q_{\text{SPG}}(f)$. This makes sense, since according to our definition in Section 3, $q_{\text{SPG}}(f)$ and $\tilde{p}_{\text{SPG}}(f)$ share the same measurable space $(\mathbb{R}^{\mathcal{T}}, \mathcal{B}_{\mathbb{R}}^{\mathcal{T}})$.

Therefore, the only difference between $q_{\text{SPG}}(f)$ and $\tilde{p}_{\text{SPG}}(f)$ is the choice of the prior distributions on $\mathbf{h}$, which is $q_\eta(\mathbf{h})$ and $p_0(\mathbf{h})$, respectively. Given this property, we can compute the KL divergence between $q_{\text{SPG}}(f)$ and $\tilde{p}_{\text{SPG}}(f)$ given *measurement points* $\mathbf{X}_n$ (Appendix A.5):

**Proposition 5** (KL divergence on measurement points between SPGs). *Let $q_{SPG}(f)$ and $\tilde{p}_{SPG}(f)$ be the SPGs defined in Equation 12 and 15. Then we have*

$$D_{KL}[q_{SPG}(\mathbf{f}^{\mathbf{X}_n})||\tilde{p}_{SPG}(\mathbf{f}^{\mathbf{X}_n})] = \mathbb{E}_{f \sim q_{SPG}(f)} \log \mathcal{Z}(\mathbf{f}^{\mathbf{X}_n}), \tag{16}$$

*where $\mathcal{Z}(\mathbf{f}^{\mathbf{X}_n})$ is the partition function, $\mathcal{Z}(\mathbf{f}^{\mathbf{X}_n}) = \int_{\mathbf{h}} \tilde{p}_{SPG}(\mathbf{h}|\mathbf{f}^{\mathbf{X}_n}) \frac{q_\eta(\mathbf{h})}{p_0(\mathbf{h})} d\mathbf{h}$.*

Note that $\mathcal{Z}(\mathbf{f}^{\mathbf{X}_n})$ is intractable to compute due to the intractability of the posterior $\tilde{p}_{\text{SPG}}(\mathbf{h}|\mathbf{f}^{\mathbf{X}_n})$. Fortunately, this is already approximated by the PoE inference net $\tilde{q}_\lambda(\mathbf{h}|\mathbf{f}^{\mathbf{X}_n})$ given by Equation 14:

$$\mathcal{Z}(\mathbf{f}^{\mathbf{X}_n}) \approx \tilde{\mathcal{Z}}(\mathbf{f}^{\mathbf{X}_n}) := \int_{\mathbf{h}} \tilde{q}_\lambda(\mathbf{h}|\mathbf{f}^{\mathbf{X}_n}) \frac{q_\eta(\mathbf{h})}{p_0(\mathbf{h})} d\mathbf{h}. \tag{17}$$

Since $q_\eta(\mathbf{h})$, $p_0(\mathbf{h})$, and $\tilde{q}_\lambda(\mathbf{h}|\mathbf{f}^{\mathbf{X}_n})$ are all Gaussian distributions, $\tilde{\mathcal{Z}}(\mathbf{f}^{\mathbf{X}_n})$ can be computed using analytic solutions. Note also that thanks to the VAE-like structure in SPGs, all the calculations are performed in the latent space, whose dimensionality is much lower than $\mathbf{f}^{\mathbf{X}}$. With the additional help of analytic solutions for $\tilde{\mathcal{Z}}(\mathbf{f}^{\mathbf{X}})$, the estimation of (17) is very efficient and scalable.

### 4.4 The final algorithm: Mini-batching and de-biasing

So far we have been discussing the estimation of the KL-divergence *on measurement points* $\mathbf{X}_n$. Next, we derive a practical algorithm based on the grid-functional ELBO $\mathcal{L}_q^{grid}$ defined in Equation 6. Applying the approximation (17) to Equation 6, our final variational objective becomes

$$\log p(\mathcal{D}) \geq \sum_i^{|\mathcal{D}|} \mathbb{E}_{q(f)}[\log p_\pi(y_i|f(\mathbf{x}_i))] - \mathbb{E}_{n, \mathbf{X}_n \sim c} D_{KL}[q(\mathbf{f}^{\mathbf{X}_n})||p(\mathbf{f}^{\mathbf{X}_n})]$$

$$\approx \sum_i^{|\mathcal{D}|} \mathbb{E}_{q(f)}[\log p_\pi(y_i|f(\mathbf{x}_i))] - \mathbb{E}_{n, \mathbf{X}_n \sim c} \mathbb{E}_{f \sim q_{\text{SPG}}(f)} \log \tilde{\mathcal{Z}}(\mathbf{f}^{\mathbf{X}_n}). \tag{18}$$

To make Equation 18 scalable to large data, we can apply mini-batch sampling to the likelihood term $\sum_i^{|\mathcal{D}|} \mathbb{E}_{q(f)}[\log p(y_i|f(\mathbf{x}_i))]$. Then the only bottleneck of Equation 18 is that the input $\mathbf{X}_n$ to the inference net $\tilde{q}_\lambda$ (used in $\tilde{\mathcal{Z}}(\mathbf{f}^{\mathbf{X}_n})$) can be very high dimensional due to the condition $\mathbf{X}_\mathcal{D} \subset \mathbf{X}_n$ required by Proposition 3. Fortunately, we can derive the following mini-batch estimators for $\mathbb{E}_{n, \mathbf{X}_n \sim c} \mathbb{E}_{f \sim q_{\text{SPG}}(f)} \log \tilde{\mathcal{Z}}(\mathbf{f}^{\mathbf{X}_n})$ (Appendix A.6 and A.7):

**Proposition 6.** $\mathbb{E}_{n, \mathbf{X}_n \sim c} \mathbb{E}_{f \sim q_{SPG}(f)} \log \tilde{\mathcal{Z}}(\mathbf{f}^{\mathbf{X}_n})$ *can be estimated by the mini-batch estimator*

$$\mathcal{J}_K := \frac{1}{2} \sum_{i=1}^H \mathbb{E}_{f \sim q_{SPG}(f)} \left[ \log \sigma_{\eta_i}^{-2} + \log \hat{\sigma}_{\lambda_i}^{-2} - \log(\sigma_{\eta_i}^{-2} + \hat{\sigma}_{\lambda_i}^{-2} - 1) - \hat{\mu}_{\lambda_i}^2 \hat{\sigma}_{\lambda_i}^{-2} - \mu_{\eta_i}^2 \sigma_{\eta_i}^{-2} \right.$$

$$\left. + (\hat{\sigma}_{\eta_i}^{-2} \hat{\mu}_{\eta_i} + \hat{\sigma}_{\lambda_i}^{-2} \hat{\mu}_{\lambda_i})^2 (\sigma_{\eta_i}^{-2} + \hat{\sigma}_{\lambda_i}^{-2} - 1)^{-1} \right], \tag{19}$$

**Algorithm 1** Functional Variational Inference (FVI)

---

**Require:** data $\mathcal{D} = \{\mathbf{x}_i, y_i\}_{i=1}^N$; prior $p(f)$; surrogate $\tilde{p}_{\text{SPG}}(f)$, variational process $q_{\text{SPG}}(f)$, likelihood function $p_\pi(y|f)$, mini-batch sizes $I$ and $K$

1: **while** not converged **do**
2:     Sample $[f_1(\cdot), f_2(\cdot), ..., f_M(\cdot)]$ from $p(f)$.
3:     Improve $\tilde{p}_{\text{SPG}}(f)$ by optimizing the aggregated ELBO in Equation 13 w.r.t. $\{\mathbf{w}_s\}, \theta, \lambda$.
4: **end while**
5: **while** not converged **do**
6:     sample mini-batch $\mathcal{I}$ from $\{1, ..., |\mathcal{D}|\}$ and a set of measure points $\mathbf{X}_n$ via c.
7:     Optimize $\hat{\mathcal{L}}_{\text{FVI}}$ in Equation 20 w.r.t. $\eta$ and $\pi$ via reparameterization tricks
8: **end while**

---

*where $H$ is the dimensionality of $\mathbf{h}$, $\mathcal{N}(\mathbf{h}; \mu_{\eta_i}, \sigma_{\eta_i}^2) = q_\eta(h_i)$, $\mathcal{N}(\mathbf{h}; \mu_{\lambda_i}, \sigma_{\lambda_i}^2) = \tilde{q}_\lambda(h_i|\mathbf{f}^{\mathbf{X}})$. $\hat{\sigma}_{\lambda_i}^{-2}$ and $\hat{\mu}_{\lambda_i}$ are the mini-batch approximators for $\mu_{\lambda_i}$ and $\sigma_{\lambda_i}^2$, respectively:*

$$\hat{\sigma}_{\lambda_i}^{-2} := \sum_{k \in \mathcal{K}} \frac{|\mathcal{D}|}{K} \sigma_{h_i|f^{\mathbf{x}_k}}^{-2} + \sum_{\mathbf{x}_l \in \mathbf{X}_n \backslash \mathbf{X}_{\mathcal{D}}} \sigma_{h_i|f^{\mathbf{x}_l}}^{-2}$$

$$\frac{\hat{\mu}_{\lambda_i}}{\hat{\sigma}_{\lambda_i}^2} := \sum_{k \in \mathcal{K}} \frac{|\mathcal{D}|}{K} \sigma_{h_i|f^{\mathbf{x}_k}}^{-2} \mu_{h_i|f^{\mathbf{x}_b}} + \sum_{\mathbf{x}_l \in \mathbf{X}_n \backslash \mathbf{X}_{\mathcal{D}}} \sigma_{h_i|f^{\mathbf{x}_l}}^{-2} \mu_{h_i|f^{\mathbf{x}_l}},$$

*where $\mathcal{K}$ is a mini-batch of size $K$ sampled from $\{1, ..., |\mathcal{D}|\}$, $\mathbf{x}_l \in \mathbf{X}_n \backslash \mathbf{X}_{\mathcal{D}}$ is a set of OOD samples sampled from $\mathcal{T}$ using c in Eq. 7, and $\mu_{h_i|f^{\mathbf{x}_k}}$ and $\sigma_{h_i|f^{\mathbf{x}_k}}^2$ are the mean and variance parameter returned from $\tilde{q}_\lambda(h_i|f(\mathbf{x}_k))$.*

The estimation in Eq. 19 is biased (but consistent). To remove the bias, we propose to debias Eq. 19 based on the Russian Roulette estimator [22]:

**Proposition 7** (Debiasing). *Let $R$ be a random integer from a distribution $\mathbb{P}(N)$ with support over the integers larger than $K$. $\mathbf{x}_0$ is a random location sampled from $\mathcal{T}$. Then $\mathbb{E}_{n, \mathbf{X}_n \sim c} \mathbb{E}_{f \sim q_{SPG}(f)} \log \tilde{\mathcal{Z}}(\mathbf{f}^{\mathbf{X}_n})$ can be estimated by $\mathbb{E}\left[ \mathcal{J}_K + \sum_{k=K}^R \frac{\Delta_k}{\mathbb{P}(N \geq k)} \right]$, where $\Delta_k = \mathcal{J}_{\mathbf{k+1}} - \mathcal{J}_{\mathbf{k}}$, and the expectation is taken over $R$, $n$, $\mathbf{X}_n$, and all mini-batches used by each $\mathcal{J}_k$ terms.*

This will enable us to also perform mini-batch sampling on the measurement points when performing FVI. Our final optimization objective function is

$$\hat{\mathcal{L}}_{\text{FVI}} := \frac{|\mathcal{D}|}{I} \sum_{i \in \mathcal{I}} \mathbb{E}_{q(f)}[\log p(y_i|f(\mathbf{x}_i))] - \mathcal{J}_K - \sum_{k=B}^R \frac{\Delta_k}{\mathbb{P}(N \geq k)}, \tag{20}$$

where $\mathcal{I}$ is a mini-batch of size $I$ for the likelihood terms, $R$ is an integer sampled from $\mathbb{P}(N)$, which is set to be $(R - K) \sim \text{Geom}(0.5)$. Finally, the full algorithm is sketched in Algorithm 1. We call this proposed method Functional Variational Inference (FVI).

**Scalability**. Our method is empirically much faster than f-BNN (Appendix C.3). When estimating $\mathcal{D}_{grid}[q(f)||p(f)]$, our method scales as $\mathcal{O}(M_q)$, where $M_q$ is the number of samples sampled from $q_{SPG}(f)$ that are used in $\mathcal{J}_K$. In practice, we use $M_q = 1$. On the contrary, the SSGE estimator used in f-BNN scales as $\mathcal{O}(M_q^3 + M_q^2 D)$, where much larger value needs to used (e.g., $M_q = 100$).

## 5 Related Works

Since BNNs and VIPs are discussed in Section 1 and 2, here we only address the rest of the related works including f-BNNs, functional priors, and Bayesian non-parameterics.

**F-BNNs and F-PO.** The functional BNN [47] is proposed to address the issue of specifying meaningful priors to BNN weights. It matches a BNN to a GP prior by minimizing the functional KL divergence estimated by score function estimators [27]. As discussed in [5], this objective is not well-defined for a wide class of distributions. Also, the score function estimators used in f-BNNs

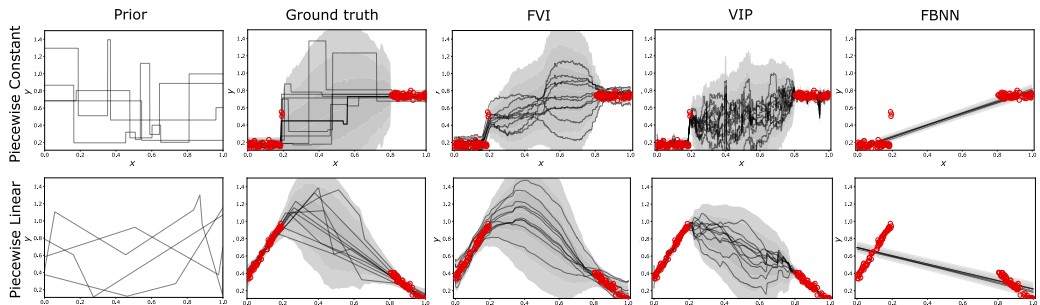

Figure 3: Implicit function prior and posterior samples from ground truth, FVI, VIP, and f-BNN, respectively. **The first row** corresponds to a piecewise constant prior, and **the second row** corresponds to a piecewise linear prior. **The leftmost column** shows 5 prior samples. **From the second column to the rightmost column** we show posterior samples generated by ground truth (returned by SIR), FVI, VIP and f-BNN, respectively. Red dots denote the training data. We plot 10 posterior samples in black lines and show predictive uncertainty as grey shaded areas.

often perform poorly in high dimensional spaces and are less efficient. In practice, we found that the f-BNN computational time is prohibitive. On the contrary, our FVI estimator avoids these issues by making use of the grid-functional divergence, which can be efficiently estimated using the latent representation of SPGs. More recently, the concurrent work of [39] proposes a tractable function space VI method for BNNs, in which the functional KL divergence is approximated via linearization. In their problem setting, the functional KL divergence is well defined since both $p$ and $q$ are BNNs with the same neural network structures. Similarly, f-POVI [50] performs inference on BNN priors by performing particle optimization in function space. One limitation of f-POVI is that it requires the prior to be reparameterizable and differentiable while our method does not have this issue.

**Function space priors.** Another line of work directly defines distributions over functions by combining stochastic processes with neural networks. For example, neural processes (NPs) [13] and its variants [24, 15] focus on meta-learning scenarios and propose to use set encoders to model all possible posterior distributions of the form $\{p(\mathbf{f}|\mathcal{C})|\mathcal{C} \subset \mathcal{D}\}$, where $\mathcal{C}$ is the so-called "context points" in neural processes. This could be inefficient for large datasets, since it needs to feed all data points to the set encoder, which scales linearly w.r.t. the dataset size. More importantly, NPs still use for learning and inference an ELBO defined on parameter space instead of function space. On the contrary, our method focuses on the functional VI for supervised learning scenarios and does not need to model all possible conditionals. When computing predictive distribution , we only need to evaluate $q_\eta(\mathbf{h})$, which is a simple Gaussian distribution (no set encoders involved).

**Bayesian non-parametrics.** In the field of Bayesian non-parameterics, Gaussian Processes (GPs) [36] and deep GPs [8] are great examples of using function-space priors to produce models that are reliable under uncertainty. To reduce the prohibitive computation cost of exact/deep GPs, various VI methods [49, 17, 32, 40] have been studied. These methods share a similar principle with our work, that is, to minimize the functional divergence between the posterior and variational processes. Nevertheless, the GP components of the functional prior play a critical role in this line of work, which makes them less applicable to general non-GP based priors.

# 6 Experiments

In this section, we evaluate the performance of FVI using a number of tasks, including interpolation with structured implicit priors, multivariate regression with BNN priors, contextual bandits, and image classification. We mainly compare FVI with other weight-space and function-space Bayesian inference methods *using the same priors*. For more implementation details, please refer to Appendix B. Additional experiments can be found in Appendix C.

## 6.1 Interpolation with non-Gaussian priors: structured implicit priors

An advantage of FVI is that it can be applied to implicit (and non-Gaussian) priors over functions, where typical GPs do not apply. In this experiment, we evaluate the interpolation task as [47]. We

Table 1: Regression experiment: Average test negative log likelihood

| DATASET | N | D | FVI | VIP-BNN | VIP-NP | BBB | VDO | $\alpha = 0.5$ | fBNN | EXACT GP |
|---|---|---|---|---|---|---|---|---|---|---|
| BOSTON | 506 | 13 | **2.33**±**0.04** | 2.45±0.04 | 2.45±0.03 | 2.76±0.04 | 2.63±0.10 | 2.45±0.02 | 2.30±0.10 | 2.46±0.04 |
| CONCRETE | 1030 | 8 | **2.88**±**0.06** | 3.02±0.02 | 3.13±0.02 | 3.28±0.01 | 3.23±0.01 | 3.06±0.03 | 3.09±0.01 | 3.05±0.02 |
| ENERGY | 768 | 8 | 0.58±0.05 | **0.56**±**0.04** | 0.60±0.03 | 2.17±0.02 | 1.13±0.02 | 0.95±0.09 | 0.68±0.02 | 0.54±0.02 |
| KIN8NM | 8192 | 8 | **-1.15**±**0.01** | -1.12±0.01 | -1.05±0.00 | -0.81±0.01 | -0.83±0.01 | -0.92±0.02 | N/A±0.00 | N/A±0.00 |
| POWER | 9568 | 4 | **2.69**±**0.00** | 2.92±0.00 | 2.90±0.00 | 2.83±0.01 | 2.88±0.00 | 2.81±0.00 | N/A±0.00 | N/A±0.00 |
| PROTEIN | 45730 | 9 | **2.85**±**0.00** | 2.87±0.00 | 2.96±0.02 | 3.00±0.00 | 2.99±0.00 | 2.90±0.00 | N/A±0.00 | N/A±0.00 |
| RED WINE | 1588 | 11 | **0.97**±**0.06** | **0.97**±**0.02** | 1.20±0.04 | 1.01±0.02 | **0.97**±**0.02** | 1.01±0.02 | 1.04±0.01 | 0.26±0.03 |
| YACHT | 308 | 6 | 0.59±0.11 | **-0.02**±**0.07** | 0.59±0.13 | 1.11±0.04 | 1.22±0.18 | 0.79±0.11 | 1.03±0.03 | 0.10±0.05 |
| NAVAL | 11934 | 16 | **-7.21**±**0.06** | -5.62±0.04 | -4.11±0.00 | -2.80±0.00 | -2.80±0.00 | -2.97±0.14 | -7.13±0.02 | N/A±0.00 |
| MEAN RANK | N/A | N/A | 1.33 | 2.11 | 3.56 | 5.22 | 4.56 | 3.33 | N/A | N/A |

Table 2: Contextual bandits performance comparison. Results are relative to the cumulative regret of the worst algorithm on each dataset. Numbers after the algorithm are the network sizes. The best methods are boldfaced, and the second best methods are highlighted in brown.

| | MEAN RANK | MUSHROOM | STATLOG | COVERTYPE | JESTER | ADULT | WHEEL | CENSUS |
|---|---|---|---|---|---|---|---|---|
| FVI $2 \times 50$ | **2.11** | 16.46 ± 2.04 | **7.95 ± 2.92** | **49.59 ± 1.61** | **68.59 ± 6.87** | **90.33 ± 0.86** | 41.44 ± 9.28 | 51.77 ± 3.06 |
| UNIFORM | 10.45 | 100.0 ± 0.00 | 99.85 ± 0.36 | 99.49 ± 0.62 | 100.0 ± 0.00 | 99.60 ± 0.53 | 94.04 ± 11.9 | 99.30 ± 0.55 |
| RMS | 5.68 | 17.74 ± 7.65 | 10.36 ± 2.51 | 69.72 ± 7.23 | 75.07 ± 5.50 | 97.65 ± 1.48 | 70.39 ± 19.7 | 94.55 ± 3.60 |
| DROPOUT $2 \times 50$ | 5.54 | 19.84 ± 6.46 | 15.53 ± 4.50 | 67.72 ± 2.32 | 75.04 ± 4.66 | 97.44 ± 0.98 | 59.40 ± 10.8 | 86.60 ± 0.52 |
| BBB $2 \times 50$ | 4.88 | 23.18 ± 5.90 | 30.90 ± 3.29 | 63.91 ± 1.96 | 72.93 ± 5.69 | 95.49 ± 2.03 | 56.38 ± 11.3 | 70.68 ± 2.32 |
| BBB $1 \times 50$ | 8.22 | **15.52 ± 4.40** | 80.25 ± 18.6 | 94.80 ± 4.84 | 83.30 ± 5.26 | 99.24 ± 0.66 | 58.12 ± 18.0 | 99.46 ± 0.37 |
| NEURALLINEAR | 6.94 | 19.04 ± 2.96 | 21.22 ± 1.98 | 75.34 ± 1.00 | 86.86 ± 3.61 | 97.93 ± 1.37 | **37.41 ± 8.86** | 83.75 ± 1.44 |
| BOOTRMS | 4.51 | 17.11 ± 5.99 | 9.47 ± 2.03 | 63.27 ± 1.35 | 74.66 ± 3.87 | 96.11 ± 1.02 | 63.15 ± 25.9 | 90.47 ± 3.40 |
| PARAMNOISE | 5.94 | 17.76 ± 4.14 | 20.95 ± 3.07 | 78.08 ± 5.66 | 76.95 ± 5.84 | 96.23 ± 1.81 | 41.26 ± 6.48 | 96.34 ± 4.56 |
| BB$\alpha$ $2 \times 50$ | 9.45 | 68.45 ± 6.05 | 95.22 ± 4.88 | 98.60 ± 1.45 | 94.29 ± 2.69 | 98.72 ± 1.28 | 80.50 ± 7.96 | 97.94 ± 2.01 |
| FBNN $2 \times 50$ | 3.17 | 16.55 ± 2.41 | 10.01 ± 1.39 | 50.10 ± 5.70 | 70.82 ± 3.27 | 90.72 ± 3.18 | 77.70 ± 21.2 | **51.22 ± 2.55** |

consider two 1-D implicit priors on $[0, 1]$: 1), piecewise constant random functions and 2), piecewise linear random functions. Please refer to Appendix B.2 for details. For each prior, we first sample a random function from the prior; then, 100 observed data points are sampled as $\mathcal{D}$, half of which are sampled from $[0, 0.2]$ and the other half are sampled from $[0.8, 1]$. Finally, we ask the algorithms to perform inference using the prior, i.e., producing samples from $p(f|\mathcal{D})$.

We compare the performance of FVI with ground truth, f-BNN and VIPs. The ground truth posterior samples are generated by sampling importance re-sampling. F-BNNs are based on the code kindly open-sourced by [47]. As we found that the training time required by f-BNN is prohibitive, we only trained f-BNN for 100 epochs for fairness (additional results for fully trained f-BNNs are provided in Appendix C.5). For VIPs (Gaussian approximations), we use an empirical covariance kernel, which is estimated from random function samples of the implicit priors. For FVI, implementation details can be found in Appendix B.2.

Results are displayed in Figure 3. FVI can successfully generate samples that mimic the piecewise constant/linear behaviors. The posterior uncertainty returned by FVI is also close to the ground truth estimates. On the other hand, f-BNNs severely under-fit the data and provide very poor in-between uncertainties. Note that, although f-BNNs are only trained for 100 epochs, their running time is still 100x higher than that of FVI (Appendix C.3). VIP performs better than f-BNNs, but fails to mimic the behaviour of the priors: the posterior samples from VIP are very noisy. This is due to the prior function samples violating the Gaussian assumption, with the correlation level between points being lower than expected. This results in very noisy VIP posterior samples that are hard to interpret.

## 6.2 Multivariate regression with BNNs priors

In this experiment, we test if the proposed FVI can perform accurate posterior inference with BNNs as functional priors. We consider multivariate regression tasks based on 9 different UCI datasets. We mainly compare with the following weight-space VI baselines for BNNs: Bayes-by-Backprop [4], variational dropout [12], and variational alpha dropout [26] ($\alpha = 0.5$). We also compare with three function-space BNN inference methods: VIP-BNNs, VIP-Neural processes [28], and f-BNNs. Finally, we include comparisons to function space particle optimization [50] in Appendix C.7 for reference purpose. All inference methods are based on the same BNN priors whenever applicable. For experimental settings, we follow [28]. Each dataset was randomly split into train (90%) and test sets (10%). This was repeated 10 times and results were averaged.

Results are shown in Table 1. Overall, FVI consistently outperforms other VI-based inference methods for BNNs and achieves the best result in 7 datasets (out of 9). FVI also outperforms f-BNNs (in 5 datasets out of 6), despite the fact that they are more expensive to train. Note that exact GPs and

Table 3: Image classification and OOD detection performance. Accuracy, negative log-likelihood (NLL) and area-under-the-curve (AUC) of OOD detection are reported. Our method outperforms all baselines in terms of classification accuracy and OOD-AUC and performs competitively on NLL for CIFAR10. Results for MAP, KFAC and Ritter et al. are obtained from [21].

| | FMNIST | | | CIFAR10 | | |
|---|---|---|---|---|---|---|
| **Model** | Accuracy | NLL | OOD-AUC | Accuracy | NLL | OOD-AUC |
| FVI | **91.60±0.14** | **0.254±0.05** | **0.956±0.06** | **77.69 ±0.64** | 0.675±0.03 | **0.883±0.04** |
| MFVI | 91.20±0.10 | 0.343±0.01 | 0.782±0.02 | 76.40±0.52 | 1.372±0.02 | 0.589±0.01 |
| MAP | 91.39±0.11 | 0.258±0.00 | 0.864±0.00 | 77.41±0.06 | 0.690±0.00 | 0.809±0.01 |
| KFAC-Laplace | 84.42±0.12 | 0.942±0.01 | 0.945±0.00 | 72.49±0.20 | 1.274±0.01 | 0.548±0.01 |
| Ritter et al. | 91.20±0.07 | 0.265±0.00 | 0.947±0.00 | 77.38±0.06 | **0.661±0.00** | 0.796±0.00 |

f-BNNs are not directly comparable to other methods, since i), they perform inference over different priors; and ii), they are much more expensive as they require the evaluation of the *exact* GP likelihood. Thus, their results are only available for smaller datasets, and are not included for ranking.

### 6.3 Contextual Bandits

Uncertainty estimates are important for downstream decision-making scenarios, since exploration-exploitation is a common dilemma that must be addressed. In this section, we consider a classic task called contextual bandits, where the agent is asked to make decisions that maximize the reward given some contexts (inputs). For this, Thompson sampling [48] is an elegant approach to guide exploration, where a model configuration is first sampled from the posterior and then an optimal action is choosen based on the sampled configuration.

We compare FVI with several NN-related baselines on datasets benchmarked by [37]. The hyperparameter settings are consistent with [47] except that we used a smaller batchsize (32). The learning rates for each baseline are tuned from $[0.001, 0.05]$. We report the cumulative regret as well as the mean ranks. Experiments are repeated for 10 runs. As shown in Table 2, no single algorithm always outperforms the others in all bandit problems. However, FVI tends to give better performance than the baselines (ranks the first overall, performs the best on 4 out of 7 datasets, and ranked top 2 on 6 out of 7 datasets), indicating that FVI can provide reliable uncertainty estimates for decision making. Moreover, FVI is much more efficient than f-BNN (nearly 500 times faster, c.f. Appendix C.4).

### 6.4 Image classification and out-of-distribution detection

To demonstrate the scalability of our method to higher dimensional data, we consider image classification tasks on Fashion MINIST [54] and CIFAR10 [25] with BNN priors. We compare our method to the following baselines: mean field VI (MFVI), maximum a posteriori (MAP), KFAC Laplace-GNN approximation [31] and its dampened version [38]. For all models, we use Bayesian CNNs with the same mixed CNN-fully connected structure as in [21, 43]. Apart from test accuracy and negative log likelihood (NLL), we also perform out-of-distribution detection using in-distribution (ID) / out-of-distribution (OOD) pairs including FashionMNIST/MNIST and CIFAR10/SVNH. Following the settings of [35, 21], we calculate the area under the curve (AUC) of out-of-distribution detection based on predictive entropies. Results are shown in Table 3. On both datasets, our proposed FVI method consistently outperforms all baselines in terms of (in-distribution) classification accuracy and OOD detection AUC. Although FVI does not achieve the best NLL on CIFAR10, it still performs competitively to MAP and dampened KFAC. This demonstrates that our method is able to scale to high dimensional data and produce accurate predictions with well-calibrated uncertainties.

## 7 Conclusion

In this paper, we propose Functional Variational Inference (FVI), a new VI method in function space. It optimizes a grid-based functional divergence, which can be estimated based on our proposed SPG model. We demonstrated that FVI works well with implicit priors, scales well to high dimensional data and provides reliable uncertainty estimates. Possible directions for future work might include developing grid-function KL estimation methods without surrogate models and improving the theoretical understanding of functional space VI.

## Acknowledgements

We thank Javier Antoran and Yingzhen Li for helpful discussions and comments during the early stage of this work.

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
