# Appendix

## A    Proof of Theoretical results

### A.1    Proof of Proposition 1 and 3

To prove Proposition 1, we first need the following lemma:

**Lemma 1** (Alternative equivalent definition of functional KL divergence [47])**.** *The KL-divergence between two stochastic processes can be estimated by the supremum of marginal KL divergences over all finite subset of inputs:*

$$D_{KL}[q(f)||p(f)] = \sup_{n,\mathbf{X}_n} D_{KL}[q(\mathbf{f}^{\mathbf{X}_n})||p(\mathbf{f}^{\mathbf{X}_n})], \tag{21}$$

*where $\mathbf{X}_n$ is the so called measurement points, $\mathbf{f}^{\mathbf{X}_n}$ is the vector of function values evaluated on $\mathbf{X}_n$, and $D_{KL}[q(\mathbf{f}^{\mathbf{X}_n})||p(\mathbf{f}^{\mathbf{X}_n})]$ is the KL-divergence over random vectors typically used in machine learning community.*

Readers may refer to [47] for the proof of this lemma.

**Proposition 1.** *Suppose $c$ has full support on $\mathcal{T}^{\mathbb{Z}^+}$. Then, $\mathcal{D}_{grid}[q(f)||p(f|\mathcal{D})]$ Satisfies the following conditions:*

- $\mathcal{D}_{grid}[q(f)||p(f|\mathcal{D})] \geq 0$
- $\mathcal{D}_{grid}[q(f)||p(f|\mathcal{D})] = 0$ *if and only if $q(f) = p(f|\mathcal{D})$*

**Proof:**    First, according the the definition of

$$\mathcal{D}_{grid}[q(f)||p(f|\mathcal{D})] = \mathbb{E}_{n,\mathbf{X}_n \sim c} D_{KL}[q(\mathbf{f}^{\mathbf{X}_n})||p(\mathbf{f}^{\mathbf{X}_n}|\mathcal{D})],$$

the positivity property holds since $D_{KL}[q(\mathbf{f}^{\mathbf{X}_n})||p(\mathbf{f}^{\mathbf{X}_n}|\mathcal{D})] \geq 0$.

Next, to prove $\mathcal{D}_{grid}[q(f)||p(f|\mathcal{D})] = 0$ if and only if $q(f) = p(f|\mathcal{D})$, we first show that

$$\arg\min_{q(f)} D_{KL}[q(f)||p(f|\mathcal{D})] = \arg\min_{q(f)} \mathbb{E}_{n,\mathbf{X}_n \sim c} D_{KL}[q(\mathbf{f}^{\mathbf{X}_n})||p(\mathbf{f}^{\mathbf{X}_n}|\mathcal{D})].$$

Let's first consider the left handside, $\arg\min_{q(f)} D_{KL}[q(f)||p(f|\mathcal{D})]$. When it reaches the optimum, we have a unique solution, $q_L^\star(f) = p(f|\mathcal{D})$. According to Equation 21, we have:

$$\arg\min_{q(f)} \sup_{n,\mathbf{X}_n} D_{KL}[q(\mathbf{f}^{\mathbf{X}_n})||p(\mathbf{f}^{\mathbf{X}_n}|\mathcal{D})] = \arg\min_{q(f)} D_{KL}[q(f)||p(f|\mathcal{D})] = q_L^\star(f).$$

Also, notice that

$$\mathbb{E}_{n,\mathbf{X}_n \sim c} D_{KL}[q(\mathbf{f}^{\mathbf{X}_n})||p(\mathbf{f}^{\mathbf{X}_n}|\mathcal{D})] \leq \sup_{n,\mathbf{X}_n} D_{KL}[q(\mathbf{f}^{\mathbf{X}_n})||p(\mathbf{f}^{\mathbf{X}_n}|\mathcal{D})]$$

At $q_L^\star(f)$, we have

$$0 \leq \mathbb{E}_{n,\mathbf{X}_n \sim c} D_{KL}[q_L^\star(\mathbf{f}^{\mathbf{X}_n})||p(\mathbf{f}^{\mathbf{X}_n}|\mathcal{D})] \leq \sup_{n,\mathbf{X}_n} D_{KL}[q_L^\star(\mathbf{f}^{\mathbf{X}_n})||p(\mathbf{f}^{\mathbf{X}_n}|\mathcal{D})] = 0$$

Therefore, we have

$$q_L^\star(f) \in \arg\min_{q(f)} \mathbb{E}_{n,\mathbf{X}_n \sim c} D_{KL}[q(\mathbf{f}^{\mathbf{X}_n})||p(\mathbf{f}^{\mathbf{X}_n}|\mathcal{D})]$$

On the other hand, assume that $\mathbb{E}_{n,\mathbf{X}_n \sim c} D_{KL}[q_L^\star(\mathbf{f}^{\mathbf{X}_n})||p(\mathbf{f}^{\mathbf{X}_n}|\mathcal{D})]$ reaches its optimum 0 at some optimal solution $q_R^\star(f)$. Since $D_{KL}[q_R^\star(\mathbf{f}^{\mathbf{X}_n})||p(\mathbf{f}^{\mathbf{X}_n}|\mathcal{D})]$ is non-negative and $c$ has full support, we have $D_{KL}[q_R^\star(\mathbf{f}^{\mathbf{X}_n})||p(\mathbf{f}^{\mathbf{X}_n}|\mathcal{D})] = 0$ for all possible $\mathbf{X}_n \subset \text{supp}(c) = \mathcal{T}^{\mathbb{Z}^+}$. Therefore, we have

$$D_{KL}[q_R^\star(f)||p(f|\mathcal{D})] = \sup_{n,\mathbf{X}_n} D_{KL}[q_R^\star(\mathbf{f}^{\mathbf{X}_n})||p(\mathbf{f}^{\mathbf{X}_n}|\mathcal{D})] = 0$$

Therefore, we have

$$q_R^\star(f) = p(f|\mathcal{D}) = q_L^\star(f)$$

That is,

$$\arg\min_{q(f)} D_{KL}[q(f)||p(f|\mathcal{D})] = \arg\min_{q(f)} \mathbb{E}_{n,\mathbf{X}_n \sim c} D_{KL}[q(\mathbf{f}^{\mathbf{X}_n})||p(\mathbf{f}^{\mathbf{X}_n}|\mathcal{D})] = p(f|\mathcal{D})$$

In other words, both the functional KL divergence and grid-functional KL divergence have the same unique global optimal solution, $q(f|\mathcal{D})$. At $q(f|\mathcal{D})$, both divergence achieves minimum value, 0. Therefore, $D_{KL}[q(f)||p(f|\mathcal{D})] = 0$ implies that $q(f)$ must be $p(f|\mathcal{D})$.

$\square$

**Proposition 3.** *Let $n, \mathbf{X}_n \sim c$ be a set of random measure points such that $\mathbf{X}_n$ always contains $\mathbf{X}_{\mathcal{D}}$. Define:*

$$\mathcal{L}_q^{grid} := \log p(\mathcal{D}) - \mathcal{D}_{grid}[q(f)||p(f|\mathcal{D})]. \tag{22}$$

*Then we have:*

$$\mathcal{L}_q^{grid} = \mathbb{E}_{q(f)}[\log p(\mathcal{D}|f)] - \mathcal{D}_{grid}[q(f)||p(f)] \tag{23}$$

*and $\log p(\mathcal{D}) \geq \mathcal{L}_q^{grid} \geq \mathcal{L}_q^{functional}$.*

**Proof:** Since $\mathcal{D}_{grid} \geq 0$, the the statement $\log p(\mathcal{D}) \geq \mathcal{L}_q^{grid}$ obviously holds. Then, notice that:

$$\mathcal{L}_q^{grid}$$
$$= \log p(\mathcal{D}) - \mathbb{E}_{n,\mathbf{X}_n \sim c} D_{KL}[q(\mathbf{f}^{\mathbf{X}_n})||p(\mathbf{f}^{\mathbf{X}_n}|\mathcal{D})]$$
$$= \mathbb{E}_{n,\mathbf{X}_n \sim c}\{\log p(\mathcal{D}) - D_{KL}[q(\mathbf{f}^{\mathbf{X}_n})||p(\mathbf{f}^{\mathbf{X}_n}|\mathcal{D})]\}$$
$$= \mathbb{E}_{n,\mathbf{X}_n \sim c}\{\log p(\mathcal{D}) - \mathbb{E}_q[\log \frac{q(\mathbf{f}^{\mathbf{X}_n})}{p(\mathbf{f}^{\mathbf{X}_n}|\mathcal{D})}]\}$$
$$= \mathbb{E}_{n,\mathbf{X}_n \sim c}\{\log p(\mathcal{D}) - \mathbb{E}_q[\log \frac{q(\mathbf{f}^{\mathbf{X}_n})p(\mathcal{D})}{p(\mathbf{f}^{\mathbf{X}_n}, \mathcal{D})}]\}$$
$$= \mathbb{E}_{n,\mathbf{X}_n \sim c}\{\mathbb{E}_q[-\log q(\mathbf{f}^{\mathbf{X}_n}) + \log p(\mathbf{f}^{\mathbf{X}_n}, \mathcal{D})]\}$$
$$= \mathbb{E}_{n,\mathbf{X}_n \sim c}\{\mathbb{E}_{q(\mathbf{f}^{\mathcal{D}})} \log p(\mathcal{D}|\mathbf{f}^{\mathcal{D}}) - D_{KL}[q(\mathbf{f}^{\mathbf{X}_n})||p(\mathbf{f}^{\mathbf{X}_n})]\}$$
$$= \mathbb{E}_{q(f)}[\log p_\pi(\mathcal{D}|f)] - \mathbb{E}_{n,\mathbf{X}_n \sim c} D_{KL}[q(\mathbf{f}^{\mathbf{X}_n})||p(\mathbf{f}^{\mathbf{X}_n})]$$

This proves the statement that $\mathcal{L}_q^{grid} = \mathbb{E}_{q(f)}[\log p_\pi(\mathcal{D}|f)] - \mathcal{D}_{grid}[q(f)||p(f)]$.

Finally, since

$$\mathbb{E}_{n,\mathbf{X}_n \sim c} D_{KL}[q(\mathbf{f}^{\mathbf{X}_n})||p(\mathbf{f}^{\mathbf{X}_n})]$$
$$\leq \mathbb{E}_{n,\mathbf{X}_n \sim c} \sup_{n,\mathbf{X}_n} D_{KL}[q(\mathbf{f}^{\mathbf{X}_n})||p(\mathbf{f}^{\mathbf{X}_n})]$$
$$= \sup_{n,\mathbf{X}_n} D_{KL}[q(\mathbf{f}^{\mathbf{X}_n})||p(\mathbf{f}^{\mathbf{X}_n})]$$
$$= D_{KL}[q(f)||p(f)],$$

Therefore we also have:

$$\mathbb{E}_{q(f)}[\log p_\pi(\mathcal{D}|f)] - \mathbb{E}_{n,\mathbf{X}_n \sim c} D_{KL}[q(\mathbf{f}^{\mathbf{X}_n})||p(\mathbf{f}^{\mathbf{X}_n})] \geq \mathcal{L}_q^{functional},$$

which concludes the first part of the proposition.

$\square$

## A.2   Proof of Proposition 2

**Proposition 2.** *Let $p(f)$ and $q(f)$ be two distributions for random functions. Assume that $p(f)$ is parameterized by the following sampling processes:*

$$f = h + \epsilon, h(\mathbf{x}) \sim p(h|\mathbf{x}; \Theta), \Theta \sim p(\Theta), \epsilon \sim \mathcal{N}(0, \sigma^2)$$

*, And $q(f)$ is parameterized by:*

$$f = h + \epsilon, h(\mathbf{x}) \sim q(h|\mathbf{x}; \Gamma), \Gamma \sim q(\Gamma), \eta \sim \mathcal{N}(0, \sigma^2).$$

*Here,$\mathbf{x} \in \mathcal{T} \subset \mathbb{R}^d$, $h$ is the random latent function, $\Theta \in \mathbb{R}^I$, $\Gamma \in \mathbb{R}^J$ are the parameters of each random function distributions, respectively. Suppose $p(h|\mathbf{x}; \Theta)$, $q(h|\mathbf{x}; \Theta)$, $p(\Theta)$ and $q(\Gamma)$ all have compact supports w.r.t. $h$, $h$, $\Theta$, and $\Gamma$, respectively (and their supports are denoted by $\mathcal{A}$, $\mathcal{B}$, $\mathcal{W}$, and $\mathcal{V}$ ). Then, there exist a sampling distribution $c$ such that: 1, $c$ has full support on $\mathcal{T}^{\mathbb{Z}^+}$, and 2, $\mathcal{D}_{grid}[q(f)||p(f)]$ is finite.*

**Proof** : Let $\mathbf{X}_n$ denote a set of $n$ measure points $\{\mathbf{x}_k\}_{1 \le k \le n}$ in $\mathcal{T}^n$. Also, let the sampling distribution $c$ to have the following form:

$$n \sim p(n), \mathbf{x}_k \sim \mathcal{U}(\mathcal{T}), \ \ \forall 1 \le k \le n$$

That is, $c$ first samples a positive integer $n$ from the distribution $p(n)$, and then draw $n$ samples from $\mathcal{T}$ independently and uniformly. Based on the choice of $p(h|\mathbf{x}; \Theta)$ and $q(h|\mathbf{x}; \Theta)$, we have two possible cases: 1), $p(h|\mathbf{x}; \Theta)$, $q(h|\mathbf{x}; \Theta)$ are stochastic; 2) $p(h|\mathbf{x}; \Theta)$ and $q(h|\mathbf{x}; \Theta)$ are deterministic, i.e., $p(h|\mathbf{x}; \Theta) = \delta(h - g(\mathbf{x}, \Theta))$ and $q(h|\mathbf{x}; \Theta) = \delta(h - v(\mathbf{x}, \Theta))$ for some functions $g$ and $v$. We will now discuss these two cases separately.

**Case 1:** $p(h|\mathbf{x}; \Theta)$ **are** $q(h|\mathbf{x}; \Theta)$ **are non-deterministic** In this case, let us consider $\mathcal{D}_{grid}[q(f)||p(f)]$. According to the definition

$$
\begin{aligned}
&\mathcal{D}_{grid}[q(f)||p(f)] \\
=&\mathbb{E}_{n, \mathbf{X}_n \sim c} D_{KL}[q(\mathbf{f}^{\mathbf{X}_n})||p(\mathbf{f}^{\mathbf{X}_n}|\mathcal{D})] \\
=&\mathbb{E}_{n \sim p(n)} \mathbb{E}_{\mathbf{X}_n \sim U(\mathcal{T}^n)} D_{KL}[q(\mathbf{f}^{\mathbf{X}_n})||p(\mathbf{f}^{\mathbf{X}_n})] \\
=&\sum_{n=1}^{\infty} p(n) \mathbb{E}_{\mathbf{X}_n \sim U(\mathcal{T}^n)} D_{KL}[q(\mathbf{f}^{\mathbf{X}_n})||p(\mathbf{f}^{\mathbf{X}_n})]
\end{aligned}
$$

Therefore, we only need to show that the series $\sum_{n=1}^{\infty} p(n) \mathbb{E}_{\mathbf{X}_n \sim U(\mathcal{T}^n)} D_{KL}[q(\mathbf{f}^{\mathbf{X}_n})||p(\mathbf{f}^{\mathbf{X}_n})]$ converges. Notice that

$$
\begin{aligned}
&\mathbb{E}_{\mathbf{X}_n \sim U(\mathcal{T}^n)} D_{KL}[q(\mathbf{f}^{\mathbf{X}_n})||p(\mathbf{f}^{\mathbf{X}_n})] \\
\le&\mathbb{E}_{\mathbf{X}_n \sim U(\mathcal{T}^n)} D_{KL}[q(\mathbf{h}^{\mathbf{X}_n})||p(\mathbf{h}^{\mathbf{X}_n})] \\
=&\mathbb{E}_{\mathbf{X}_n \sim U(\mathcal{T}^n)} \int_{\mathbf{h}^{\mathbf{X}_n}} p(\mathbf{h}^{\mathbf{X}_n}) \log \frac{p(\mathbf{h}^{\mathbf{X}_n})}{q(\mathbf{h}^{\mathbf{X}_n})} d\mathbf{h}^{\mathbf{X}_n} \\
\le&\mathbb{E}_{\mathbf{X}_n \sim U(\mathcal{T}^n)} \left[ \log \bar{p} - \log \underline{q} \right] \\
\le& \sup_{\mathbf{X}_n \in \mathcal{T}^n} \left[ \log \bar{p} - \log \underline{q} \right]
\end{aligned}
\tag{24}
$$

The first inequality is due to information processing inequality. The $\bar{p}$ and $\underline{q}$ in the second inequality is defined as

$$\bar{p} = \sup_{\mathbf{h}^{\mathbf{X}_n} \in \mathcal{A}^n \subset \mathbb{R}^n} p(\mathbf{h}^{\mathbf{X}_n}) > 0$$

$$\underline{q} = \inf_{\mathbf{h}^{\mathbf{X}_n} \in \mathcal{B}^n \subset \mathbb{R}^n} q(\mathbf{h}^{\mathbf{X}_n}) > 0,$$

where $\mathcal{A}^n$ and $\mathcal{B}^n$ are the compact support of $p(\mathbf{h}^{\mathbf{X}_n})$ and $q(\mathbf{h}^{\mathbf{X}_n})$, respectively (since both $p(h|\mathbf{x}; \Theta)$ and $q(h|\mathbf{x}; \Theta)$ have compact support in $\mathbb{R}$, $p(\mathbf{h}^{\mathbf{X}_n})$ and $q(\mathbf{h}^{\mathbf{X}_n})$ have compact support in $\mathbb{R}^n$ ). Note that both $\bar{p}$ and $\underline{q}$ are strictly greater than 0, due to the the the fact that $\mathcal{A}^n$ and $\mathcal{B}^n$ is the support of

$p(\mathbf{h}^{\mathbf{X}_n})$ and $q(\mathbf{h}^{\mathbf{X}_n})$, respectively. Next, notice that

$$\bar{p}$$
$$= \sup_{\mathbf{h}^{\mathbf{X}_n} \in \mathcal{A}^n} p(\mathbf{h}^{\mathbf{X}_n})$$
$$= \sup_{\mathbf{h}^{\mathbf{X}_n} \in \mathcal{A}^n} \int_{\Theta} \prod_{1 \leq k \leq n} p(h_i|\mathbf{x_i}; \Theta)p(\Theta)d\Theta$$
$$\leq \sup_{\mathbf{h}^{\mathbf{X}_n} \in \mathcal{A}^n} \sup_{\Theta \in \mathcal{W}} \sup_{\mathbf{X}_n \in \mathcal{T}^n} \prod_{1 \leq k \leq n} p(h_i|\mathbf{x_i}; \Theta)$$
$$\leq \prod_{1 \leq k \leq n} \left( \sup_{h_i \in \mathcal{A}} \sup_{\Theta \in \mathcal{W}} \sup_{\mathbf{x}_i \in \mathcal{T}} p(h_i|\mathbf{x_i}; \Theta) \right)$$
$$= (p^\star)^n > 0$$

Where we have used $(p^\star)$ to denote $\left( \sup_{h_i \in \mathcal{A}} \sup_{\Theta \in \mathcal{W}} \sup_{\mathbf{x}_i \in \mathcal{T}} p(h_i|\mathbf{x_i}; \Theta) \right)$. The existence of $(p^\star)$ is due to the compactness of $\mathcal{A}$, $\mathcal{W}$, and $\mathcal{T}$ stated in our assumptions. Note that $p^\star$ is strictly greater than 0 since $\mathcal{A}$ is the support of $p(h|\mathbf{x}; \Theta)$. Similarly, we have

$$\underline{q}$$
$$= \inf_{\mathbf{h}^{\mathbf{X}_n} \in \mathcal{B}^n} q(\mathbf{h}^{\mathbf{X}_n})$$
$$= \inf_{\mathbf{h}^{\mathbf{X}_n} \in \mathcal{B}^n} \int_{\Gamma} \prod_{1 \leq k \leq n} q(h_i|\mathbf{x_i}; \Gamma)q(\Gamma)d\Gamma$$
$$\geq \inf_{\mathbf{h}^{\mathbf{X}_n} \in \mathcal{B}^n} \inf_{\Gamma \in \mathcal{V}} \inf_{\mathbf{X}_n \in \mathcal{T}^n} \prod_{1 \leq k \leq n} q(h_i|\mathbf{x_i}; \Gamma)$$
$$\geq \prod_{1 \leq k \leq n} \left( \inf_{h_i \in \mathcal{B}} \inf_{\Gamma \in \mathcal{V}} \inf_{\mathbf{x}_i \in \mathcal{T}} q(h_i|\mathbf{x_i}; \Gamma) \right)$$
$$= (q^\star)^n > 0$$

Therefore, back to inequality 24, we have:

$$\mathbb{E}_{\mathbf{X}_n \sim U(\mathcal{T}^n)} D_{KL}[q(\mathbf{f}^{\mathbf{X}_n})||p(\mathbf{f}^{\mathbf{X}_n})]$$
$$\leq \sup_{\mathbf{X}_n \in \mathcal{T}^n} \left[ \log \bar{p} - \log \underline{q} \right]$$
$$\leq n \left[ \log p^\star - \log q^\star \right]$$

Now, let us consider the series $\sum_{n=1}^{\infty} p(n) n \left[ \log p^\star - \log q^\star \right]$. Apparently, based on d'Alembert's criterion, this series is absolute convergent if we can choose $p(n)$ such that $\lim_{n \to \infty} p(n+1)/p(n) < 1$. For example, $p(n)$ could be a geometric distribution with mean parameter greater than 1 (or success probability that is strictly greater than 0, and strictly smaller than 1).. Since geometric distribution has full support in $\mathbb{Z}^+$, it satisfies the claim of this proposition. Finally, given such $p(n)$ distribution, $\sum_{n=1}^{\infty} p(n)\mathbb{E}_{\mathbf{X}_n \sim U(\mathcal{T}^n)} D_{KL}[q(\mathbf{f}^{\mathbf{X}_n})||p(\mathbf{f}^{\mathbf{X}_n})]$ is also convergent due to direct comparison test.

**Case 2:** $p(h|\mathbf{x}; \Theta)$ **are** $q(h|\mathbf{x}; \Theta)$ **are deterministic**, i.e., $p(h|\mathbf{x}; \Theta) = \delta(h - g(\mathbf{x}, \Theta))$ and $q(h|\mathbf{x}; \Gamma) = \delta(h - v(\mathbf{x}, \Gamma))$ for some functions $g$ and $v$. Inequality 24 still holds. However, the upper bounds for $\bar{p}$ and $\underline{q}$ are different:

$$\bar{p}$$
$$= \sup_{\mathbf{h}^{\mathbf{X}_n} \in \mathcal{A}^n} p(\mathbf{h}^{\mathbf{X}_n})$$
$$= \sup_{\mathbf{h}^{\mathbf{X}_n} \in \mathcal{A}^n} \int_{\Theta \in \mathcal{W}} \mathbb{I}_{\{\Theta | \mathbf{h}^{\mathbf{X}_n} = g(\mathbf{X}; \Theta)\}} p(\Theta)d\Theta$$
$$\leq \sup_{\mathbf{h}^{\mathbf{X}_n} \in \mathcal{A}^n} L(\mathcal{W}) \sup_{\Theta \in \mathcal{W}} p(\Theta)$$
$$= (p^\star) > 0$$

Where we have used $(p^\star)$ to denote $L(\mathcal{W}) \sup_{\Theta \in \mathcal{W}} p(\Theta)$, and $L$ is the Lebesgue measure on $\mathbb{R}^I$. The existence of $(p^\star)$ is due to the compactness of $\mathcal{W}$ stated in our assumptions. $p^\star$ is strictly greater than 0 since $\mathcal{W}$ is the support of $p(\Theta)$. Similarly, we have:

$$\underline{q}$$
$$= \inf_{\mathbf{h}^{\mathbf{X}_n} \in \mathcal{A}^n} p(\mathbf{h}^{\mathbf{X}_n})$$
$$= \inf_{\mathbf{h}^{\mathbf{X}_n} \in \mathcal{A}^n} \int_{\Gamma \in \mathcal{V}} \mathbb{I}_{\{\Gamma | \mathbf{h}^{\mathbf{X}_n} = g(\mathbf{X};\Gamma)\}} p(\Gamma) d\Gamma$$
$$\geq \inf_{\mathbf{h}^{\mathbf{X}_n} \in \mathcal{A}^n} L(\mathcal{V}) \inf_{\Gamma \in \mathcal{V}} p(\Gamma)$$
$$= (q^\star) > 0$$

Therefore, back to inequality 24, we have:

$$\mathbb{E}_{\mathbf{X}_n \sim U(\mathcal{T}^n)} D_{KL}[q(\mathbf{f}^{\mathbf{X}_n}) || p(\mathbf{f}^{\mathbf{X}_n})]$$
$$\leq \sup_{\mathbf{X}_n \in \mathcal{T}^n} \left[ \log \bar{p} - \log \underline{q} \right] \bar{p}$$
$$\leq \left[ \log p^\star - \log q^\star \right] (p^\star)$$

Again, consider the series $\sum_{n=1}^{\infty} p(n) \left[ \log p^\star - \log q^\star \right] (p^\star)$. Apparently, based on d'Alembert's criterion, this series is absolute convergent if we can choose $p(n)$ such that $\lim_{n \to \infty} p(n+1)/p(n) < 1$. Similar to the first case, let $p(n)$ be a geometric distribution with mean parameter greater than 1 (or success probability that is strictly greater than 0, and strictly smaller than 1). Finally, given such $p(n)$ distribution, $\sum_{n=1}^{\infty} p(n) \mathbb{E}_{\mathbf{X}_n \sim U(\mathcal{T}^n)} D_{KL}[q(\mathbf{f}^{\mathbf{X}_n}) || p(\mathbf{f}^{\mathbf{X}_n})]$ is also convergent due to direct comparison test.

$\square$

**Remark (grid-functional KL using BNN as priors).** The compactness assumption in Proposition 2 seems restrictive, since BNNs with Gaussian priors on weights will break the compactness assumption. Indeed, the assumptions in proposition 2 are merely sufficient conditions. However, we here note that the proof still holds under BNN priors. Assume $p(h_i | \mathbf{x}_i; \Theta = \mathbf{w}) = \mathcal{N}(h_i; g_{\mathbf{w}}(\mathbf{x}_i), \varsigma^2)$, where $g_{\mathbf{w}}(\cdot)$ is a Bayesian neural network parameterized by $\mathbf{w}$, and $p(\mathbf{w})$ is some suitable prior on weights such as factorized Gaussians. In this case, it is trivial to see that $p^\star = \sup_{h_i} \sup_{\mathbf{w}} \sup_{\mathbf{x}_i} p(h_i | \mathbf{x_i}; \Theta) = \frac{1}{\sqrt{2\pi\varsigma^2}} > 0$, hence the rest of the proof still holds.

## A.3 Grid-Functional KL between a parametric model and a Gaussian process

In this section, we discuss the non-parametric counter part of Proposition 2, i.e., is the grid functional KL between a parametric model and a Gaussian process is still finite? Without loss of generality, let us consider the example of the approximate inference problem considered in variational implicit processes (VIP), where a GP is used as a variational distribution to approximate another stochastic process. To be concrete, assume $p(f)$ is a parametric model parameterized as in Proposition 1, and $q(f)$ is a zero mean Gaussian process with kernel function $K(\cdot, \cdot)$. Assume that $K(\cdot, \cdot)$ is a stationary kernel, i.e., $K(\mathbf{x}_1, \mathbf{x}_2) = \Phi(\|\mathbf{x}_1 - \mathbf{x}_2\|)$ for some function $\Phi$ (e.g., radial basis function). In fact, we have the following Corollary:

**Corollary 1.** *Let $p(f)$ and $q(f)$ be two distributions for random functions. Assume that $p(f)$ is parameterized by the following sampling processes:*

$$f = h + \epsilon, h(\mathbf{x}) \sim p(h | \mathbf{x}; \Theta), \Theta \sim p(\Theta), \epsilon \sim \mathcal{N}(0, \sigma^2),$$

*and $q(f)$ is parameterized by a zero mean Gaussian process with kernel function $K(\cdot, \cdot)$.*

*Assume further that: i), $p(f)$ satisfies the assumptions in Proposition 2; ii), $K(\cdot, \cdot)$ is a stationary kernel, i.e., $K(\mathbf{x}_1, \mathbf{x}_2) = \Phi(\|\mathbf{x}_1 - \mathbf{x}_2\|)$ for some function $\Phi$ (e.g., radial basis function). and iii), the smallest eigen value of $\mathbf{K}_{\mathbf{X}_n, \mathbf{X}_n}$, denoted by $\lambda_n$, decays in the order of $\mathcal{O}(n^{-\gamma})$ for some constant $\gamma > 1$ (see the literature of eigen value distribution/lower bounding smallest eigen value of kernel matrices, and/or norm estimation for inverse matrices. For example, [51, 2, 52, 3, 42, 33] to name a few).*

*Then, there exist a sampling distribution $c$ such that: 1, $c$ has full support on $\mathcal{T}^{\mathbb{Z}^+}$, and 2, $\mathcal{D}_{grid}[q(f)||p(f)]$ is finite.*

**Proof** We can basically apply most of the proof of Proposition 2. In our case, the key ingredient is to derive a lower bound for

$$\underline{q} = \inf_{\mathbf{h}^{\mathbf{X}_n} \in \mathcal{B}^n \subset \mathbb{R}^n} q(\mathbf{h}^{\mathbf{X}_n}).$$

Since $q(\mathbf{h}^{\mathbf{X}_n})$ is a GP as described before, its likelihood function is given by

$$\log q(\mathbf{h}^{\mathbf{X}_n}) = -\frac{\mathbf{h}^{\mathbf{X}_n T} \mathbf{K}_{\mathbf{X}_n,\mathbf{X}_n}^{-1} \mathbf{h}^{\mathbf{X}_n}}{2} - \frac{n}{2} \log 2\pi - \frac{1}{2} \log |\mathbf{K}_{\mathbf{X}_n,\mathbf{X}_n}|$$

. Without loss of generality, assume that $\|\mathbf{h}^{\mathbf{X}_n}\| \leq A$ for some constant A. Then, we have

$$\mathbf{h}^{\mathbf{X}_n T} \mathbf{K}_{\mathbf{X}_n,\mathbf{X}_n}^{-1} \mathbf{h}^{\mathbf{X}_n} \leq \frac{1}{\lambda_n} \|\mathbf{h}^{\mathbf{X}_n}\| \leq \frac{A}{\lambda_n},$$

where $\lambda_n$ denotes the smallest eigen value for $\mathbf{K}_{\mathbf{X}_n,\mathbf{X}_n}$ (or equivalently, $\frac{1}{\lambda_n}$ is the largest eigen value for $\mathbf{K}_{\mathbf{X}_n,\mathbf{X}_n}^{-1}$).

Notice also that

$$\log |\mathbf{K}_{\mathbf{X}_n,\mathbf{X}_n}| \leq n \log \frac{1}{n} \mathrm{Tr}(\mathbf{K}_{\mathbf{X}_n,\mathbf{X}_n}) = n \log \Phi(0).$$

Therefore, we can write

$$\log \underline{q} \geq -\frac{n}{2}(\log 2\pi + \log \Phi(0)) - \frac{A}{2\lambda_n}$$

By the same argument used in Proposition 2, we have

$$\mathbb{E}_{\mathbf{X}_n \sim U(\mathcal{T}^n)} D_{KL}[q(\mathbf{f}^{\mathbf{X}_n}) || p(\mathbf{f}^{\mathbf{X}_n})]$$
$$\leq n \left[ \log p^\star + \frac{1}{2}(\log 2\pi + \log \Phi(0)) \right] + \frac{A}{2\lambda_n}$$

Since $\lambda_n$ decays in the order of $\mathcal{O}(n^{-\gamma})$ for some constant $\gamma > 1$, by running the same argument as in the proof of Proposition 2, $\sum_{n=1}^{\infty} p(n) \mathbb{E}_{\mathbf{X}_n \sim U(\mathcal{T}^n)} D_{KL}[q(\mathbf{f}^{\mathbf{X}_n}) || p(\mathbf{f}^{\mathbf{X}_n})]$ is absolute convergent if $\lim_{n \to \infty} p(n+1)/p(n) < 1$. $\qquad \square$

### A.4   Proof of Proposition 4

**Proposition 4** (Expressiveness of SPGs). *Let $p(f)$ be a square-integrable stochastic process defined on probability space $(\mathcal{X}, \mathcal{B})$, and its index set $\mathcal{T}$ is a compact subset of $\mathbb{R}^d$. Here, $\mathcal{X}$ is a compact metric space, $\mathcal{B}$ is the Borel set on $\mathcal{X}$. Then, for $\forall \epsilon > 0$, there exists a SPG $q_{SPG}^\epsilon(f)$ with a Gaussian prior on latent space, such that:*

$$\mathrm{MMD}(p, q_{SPG}^\epsilon; \mathcal{F}) < \epsilon \quad for \quad \forall \mathbf{x} \in \mathcal{T},$$

*where $\mathrm{MMD}$ is the maximum mean discrepancy between $p$ and $q$, $\mathcal{F}$ is the MMD function class defined to be a unit ball in a reproducing kernel Hilbert space (RKHS) with a universal kernel [46] $k(\cdot, \cdot)$ as its reproducing kernel.*

**Proof** Since $p(f)$ is a stochastic process defined on $\mathcal{L}^2(\mathcal{T})$, we can apply Karhunen–Loeve expansion to $f(\mathbf{x})$. Specifically, we have:

$$f(\mathbf{x}) = \lim_{N \to \infty} L_N, \quad L_N := \sum_i^N Z_i \phi_i(\mathbf{x}), \quad \sum_i^\infty \lambda_i < +\infty.$$

Where the limit is in the sense of (uniform) convergence in $\mathcal{L}^2(\mathcal{T})$, $Z_i$ are zero-mean, uncorrelated random variables with variance $\lambda_i$. Here $\{\phi_i\}_{i=1}^\infty$ is an orthonormal basis of $\mathcal{L}^2(\mathbb{R}^d)$ that are also eigen functions of the operator $O_C(f)$ defined by $O_C(f)(\mathbf{x}) = \int C(\mathbf{x}, \mathbf{x}') z(\mathbf{x}') d\mathbf{x}'$, $C(\mathbf{x}, \mathbf{x}')$ is the covariance function of $f(\cdot)$. The variance $\lambda_i$ of $Z_i$ is the corresponding eigen value of $\phi_i(\mathbf{x})$.

Then, notice that since we have assumed that $k$ is universal and $\mathcal{T}$ is a compact metric space, by Theorem 23 of [45] we have that $\mathrm{MMD}(\cdot, \cdot; \mathcal{F})$ metrizes the weak convergence of probability

measures on $\mathcal{P}$, where $\mathcal{P}$ is the set of all Borel measures on $(\mathcal{X}, \mathcal{B})$. Here, "metrization" means that for any sequence of measures $\mathbb{P}_1, \mathbb{P}_2, ..., \mathbb{P}_n, ... \in \mathcal{P}$, we have

$$\mathbb{P}_n \overset{w}{\to} \mathbb{P} \Leftrightarrow \lim_{n \to \infty} \mathrm{MMD}(\mathbb{P}_n, \mathbb{P}; \mathcal{F}) = 0.$$

Since convergence of $L_N \to f$ in $\mathcal{L}^2$ implies weak convergence, we can apply this theorem to $p(f)$, and show that:

$$\lim_{n \to \infty} \mathrm{MMD}(p_{L_n}, p; \mathcal{F}) = 0$$

holds uniformly in $\mathcal{L}^2(\mathcal{T})$. Next, given a SPG $q_{\mathrm{SPG}}$, we have:

$$\mathrm{MMD}(q_{\mathrm{SPG}}, p; \mathcal{F}) \leq \mathrm{MMD}(p_{L_n}, p; \mathcal{F}) + \mathrm{MMD}(q_{\mathrm{SPG}}, p_{L_n}; \mathcal{F}), \quad \forall n \in \mathbb{Z}_+$$

The above triangle inequality holds since $k$ is universal [16]. Hence, to prove our theorem, it sufficies to show that there exits a sequence of SPGs $q_{\mathrm{SPG},1}, ..., q_{\mathrm{SPG},n'}, ...$ such that $\lim_{n' \to \infty} \mathrm{MMD}(q_{\mathrm{SPG},n'}, p_{L_n}; \mathcal{F}) = 0, \quad \forall n \in \mathbb{Z}_+, \mathbf{x} \in \mathcal{T}$. To prove this, let us fix $n$ for now, and consider the random coefficients $\{Z_i\}_{i=1}^n$ of $L_n$. Based on the results from [7], there exists a sequence of Gaussian VAEs $q_{\mathrm{VAE},1}(\{Z_i\}_{i=1}^n), ..., q_{\mathrm{VAE},n''}(\{Z_i\}_{i=1}^n), ...$ of latent size $n$, such that

$$q_{\mathrm{VAE},n''}(\{Z_i\}_{i=1}^n) \overset{w}{\to} p(\{Z_i\}_{i=1}^n).$$

Then, define our sequence of SPGs to be:

$$q_{\mathrm{SPG},n'} = \sum_i^n \tilde{Z}_i \phi_i, \quad , \{\tilde{Z}_i\}_{i=1}^n \sim q_{\mathrm{VAE},n'}(\{Z_i\}_{i=1}^n).$$

Based on our definition in Section 4.2, $q_{\mathrm{SPG},n'}$ is indeed a SPG. Since the linear summation over $\phi_i$ using linear weights $\{Z_i\}_{i=1}^n$ is a continuous mapping, we also have:

$$q_{\mathrm{SPG},n'} \overset{w}{\to} p_{L_n}, \quad \forall n \in \mathbb{Z}_+, \mathbf{x} \in \mathcal{T}$$

due to continuous mapping theorem. Again, from the MMD metrization, we have

$$\lim_{n' \to \infty} \mathrm{MMD}(q_{\mathrm{SPG},n'}, p_{L_n}; \mathcal{F}) = 0, \quad \forall n \in \mathbb{Z}_+, \mathbf{x} \in \mathcal{T}.$$

To finally prove our theorem, consider an arbitrary error $\epsilon$. Then, there exists $L_n$ such that $\mathrm{MMD}(p_{L_n}, p; \mathcal{F}) < \epsilon/2$. Next, given this particular $L_n$, there exits $n'$ such that $\mathrm{MMD}(p_{L_n}, q_{\mathrm{SPG},n'}; \mathcal{F}) < \epsilon/2$. Together, we have:

$$\mathrm{MMD}(q_{\mathrm{SPG},n'}, p; \mathcal{F}) \leq \mathrm{MMD}(p_{L_n}, p; \mathcal{F}) + \mathrm{MMD}(q_{\mathrm{SPG},n'}, p_{L_n}; \mathcal{F}) < \epsilon/2 + \epsilon/2 = \epsilon$$

which completes the proof our theorem.

$\square$

## A.5 Proof of Proposition 5

**Proposition 5** (functional KL divergence on measurement points for SPGs). *Let $q_{SPG}(f)$ and $\tilde{p}_{SPG}(f)$ be the SPGs defined in Equation 12 and 15. Then we have:*

$$D_{KL}[q_{SPG}(\mathbf{f}^{\mathbf{X}_n}) || \tilde{p}_{SPG}(\mathbf{f}^{\mathbf{X}_n})] = \mathbb{E}_{f \sim q_{SPG}(f)} \log \mathcal{Z}(\mathbf{f}^{\mathbf{X}_n}),$$

*where $\mathcal{Z}(\mathbf{f}^{\mathbf{X}_n})$ is the partition function, $\mathcal{Z}(\mathbf{f}^{\mathbf{X}_n}) = \int_{\mathbf{h}} \tilde{p}_{SPG}(\mathbf{h}|\mathbf{f}^{\mathbf{X}_n}) \frac{q_\eta(\mathbf{h})}{p_0(\mathbf{h})} d\mathbf{h}$.*

**Proof** First, we have

$$D_{KL}[q_{\mathrm{SPG}}(\mathbf{f}^{\mathbf{X}_n}) || \tilde{p}_{\mathrm{SPG}}(\mathbf{f}^{\mathbf{X}_n})]$$

$$= D_{KL}[q_\eta(\mathbf{h}) || p_0(\mathbf{h})] - \mathbb{E}_{f \sim q_{\mathrm{SPG}}(f)} D_{KL}[q_{\mathrm{SPG}}(\mathbf{h}|\mathbf{f}^{\mathbf{X}_n}) || \tilde{p}_{\mathrm{SPG}}(\mathbf{h}|\mathbf{f}^{\mathbf{X}_n})]$$

$$= D_{KL}[q_\eta(\mathbf{h}) || p_0(\mathbf{h})] - \mathbb{E}_{f \sim q_{\mathrm{SPG}}(f)} D_{KL}[\tilde{p}_{\mathrm{SPG}}(\mathbf{h}|\mathbf{f}^{\mathbf{X}_n}) \frac{q_\eta(\mathbf{h})}{\mathcal{Z}(\mathbf{f}^{\mathbf{X}_n}) p_0(\mathbf{h})} || \tilde{p}_{\mathrm{SPG}}(\mathbf{h}|\mathbf{f}^{\mathbf{X}_n})]$$

$$= D_{KL}[q_\eta(\mathbf{h}) || p_0(\mathbf{h})] - \mathbb{E}_{f \sim q_{\mathrm{SPG}}(f), \mathbf{h} \sim q_{\mathrm{SPG}}(\mathbf{h}|\mathbf{f}^{\mathbf{x}_n})} \log \frac{q_\eta(\mathbf{h})}{p_0(\mathbf{h})} + \mathbb{E}_{f \sim q_{\mathrm{SPG}}(f)} \log \mathcal{Z}(\mathbf{f}^{\mathbf{X}_n})$$

$$= \mathbb{E}_{f \sim q_{\mathrm{SPG}}(f)} \log \mathcal{Z}(\mathbf{f}^{\mathbf{X}_n})$$

where the first equality directly follows from the chain rule of KL-divergence, and the second equality follows from the fact that $q_{\mathrm{SPG}}(\mathbf{h}|\mathbf{f}^{\mathbf{X}_n})) \propto q_{\mathrm{SPG}}(\mathbf{h})\tilde{p}_{\mathrm{SPG}}(\mathbf{f}^{\mathbf{X}_n}|\mathbf{h}), \tilde{p}_{\mathrm{SPG}}(\mathbf{h}|\mathbf{f}^{\mathbf{X}_n}) \propto p_0(\mathbf{h})\tilde{p}_{\mathrm{SPG}}(\mathbf{f}^{\mathbf{X}_n}|\mathbf{h})$.

$\square$

## A.6   Proof of Proposition 6

**Proposition 6** (Biased Mini-batch estimation of log-partition function). $\mathbb{E}_{n,\mathbf{X}_n \sim c}\mathbb{E}_{f \sim q_{SPG}(f)} \log \tilde{\mathcal{Z}}(\mathbf{f}^{\mathbf{X}_n})$ *can be estimated by the following mini-batch estimator:*

$$
\begin{aligned}
\mathcal{J}_K := \frac{1}{2}\sum_{i=1}^{H} \mathbb{E}_{f \sim q_{SPG}(f)} \Big[ & \log \sigma_{\eta_i}^{-2} + \log \hat{\sigma}_{\lambda_i}^{-2} \\
& - \log(\sigma_{\eta_i}^{-2} + \hat{\sigma}_{\lambda_i}^{-2} - 1) - \hat{\mu}_{\lambda_i}^2 \hat{\sigma}_{\lambda_i}^{-2} - \mu_{\eta_i}^2 \sigma_{\eta_i}^{-2} \\
& + (\hat{\sigma}_{\eta_i}^{-2}\hat{\mu}_{\eta_i} + \hat{\sigma}_{\lambda_i}^{-2}\hat{\mu}_{\lambda_i})^2 (\sigma_{\eta_i}^{-2} + \hat{\sigma}_{\lambda_i}^{-2} - 1)^{-1} \Big],
\end{aligned}
\tag{25}
$$

*where $H$ is the dimensionality of $\mathbf{h}$, $\mathcal{N}(\mathbf{h}; \mu_{\eta_i}, \sigma_{\eta_i}^2) = q_\eta(h_i)$, $\mathcal{N}(\mathbf{h}; \mu_{\lambda_i}, \sigma_{\lambda_i}^2) = \tilde{q}_\lambda(h_i|\mathbf{f}^{\mathbf{X}_n})$. $\hat{\sigma}_{\lambda_i}^{-2}$ and $\hat{\mu}_{\lambda_i}$ are the mini-batch approximators for $\mu_{\lambda_i}$ and $\sigma_{\lambda_i}^2$, respectively:*

$$
\hat{\sigma}_{\lambda_i}^{-2} := \sum_{k \in \mathcal{K}} \frac{|\mathcal{D}|}{K} \sigma_{h_i|f^{\times k}}^{-2} + \sum_{\mathbf{x}_l \in \mathbf{X}_n \setminus \mathbf{X}_\mathcal{D}} \sigma_{h_i|f^{\times l}}^{-2}
$$

$$
\frac{\hat{\mu}_{\lambda_i}}{\hat{\sigma}_{\lambda_i}^2} := \sum_{k \in \mathcal{K}} \frac{|\mathcal{D}|}{K} \sigma_{h_i|f^{\times k}}^{-2}\mu_{h_i|f^{\times b}} + \sum_{\mathbf{x}_l \in \mathbf{X}_n \setminus \mathbf{X}_\mathcal{D}} \sigma_{h_i|f^{\times l}}^{-2}\mu_{h_i|f^{\times l}}
$$

*where $\mathcal{K}$ is a mini-batch of size $K$ sampled from $\{1, ..., |\mathcal{D}|\}$, $\mathbf{x}_l \in \mathbf{X}_n \setminus \mathbf{X}_\mathcal{D}$ is a set of OOD samples sampled from $\mathcal{T}$ using $c$ in Eq. 7, and $\mu_{h_i|f^{\times k}}$ and $\sigma_{h_i|f^{\times k}}^2$ are the mean and variance parameter returned from $\tilde{q}_\lambda(h_i|f(\mathbf{x}_k))$.*

**Proof**   To derive the mini-batch estimator, we first compute the expression for $\tilde{\mathcal{Z}}(\mathbf{f}^{\mathbf{X}_n})$. Since $\tilde{q}_\lambda(\mathbf{h}|\mathbf{f}^{\mathbf{X}_n})$ is a product of Gaussian encoder, its mean and variance can be computed by:

$$
\Sigma_\lambda^{-1} = \sum_{\mathbf{x} \in \mathbf{X}_n} \Sigma_{h_i|f^{\times}}^{-1}
$$

$$
\mu_\lambda = \Sigma_\lambda \sum_{\mathbf{x} \in \mathbf{X}_n} \Sigma_{h_i|f^{\times}}^{-1}\mu_{h_i|f^{\times}}
$$

where $\Sigma_{h_i|f^{\times}}$ is a diagonal matrix with component $(\Sigma_{h_i|f^{\times}})_{ii} = \sigma_{h_i|f^{\times}}^2$. Let $\Sigma_\eta$ and $\mu_\eta$ be the covariance and mean of $q_\eta(\mathbf{h})$. By our assumptions, $\Sigma_\eta$ is also a diagonal matrix with $(\Sigma_\eta)_{ii} = \sigma_{\eta_i}^2$. Since $\tilde{q}_\lambda(\mathbf{h}|\mathbf{f}^{\mathbf{X}_n})\frac{q_\eta(\mathbf{h})}{p_0(\mathbf{h})}$ is a product of three Gaussian distributions, its log normalization constant $\log \tilde{\mathcal{Z}}$ can be computed using the results from, for example Appendix A.2 of [18]:

$$
\begin{aligned}
& \mathbb{E}_{n,\mathbf{X}_n \sim c}\mathbb{E}_{f \sim q_{SPG}(f)} \log \tilde{\mathcal{Z}}(\mathbf{f}^{\mathbf{X}_n}) \\
=& \mathbb{E}_{n,\mathbf{X}_n \sim c}\mathbb{E}_{f \sim q_{SPG}(f)} \Big[ \frac{1}{2}\log|\Sigma_\eta^{-1}| + \frac{1}{2}\log|\Sigma_\lambda^{-1}| - \frac{1}{2}\log|\Sigma_\eta^{-1} + \Sigma_\lambda^{-1} - I| \\
& - \frac{1}{2}\mu_\lambda^T \Sigma_\lambda^{-1}\mu_\lambda - \frac{1}{2}\mu_\eta^T \Sigma_\eta^{-1}\mu_\eta + \frac{1}{2}(\Sigma_\eta^{-1}\mu_\lambda + \Sigma_\lambda^{-1}\mu_\lambda)^T(\Sigma_\eta^{-1} + \Sigma_\lambda^{-1} - I)^{-1}(\Sigma_\eta^{-1}\mu_\lambda + \Sigma_\lambda^{-1}\mu_\lambda) \Big] \\
=& \frac{1}{2}\sum_{i=1}^{H} \mathbb{E}_{n,\mathbf{X}_n \sim c}\mathbb{E}_{f \sim q_{SPG}(f)} \Big[ \log \sigma_{\eta_i}^{-2} + \log \sigma_{\lambda_i}^{-2} - \log(\sigma_{\eta_i}^{-2} + \sigma_{\lambda_i}^{-2} - 1) \\
& - \mu_{\lambda_i}^2 \sigma_{\lambda_i}^{-2} - \mu_{\eta_i}^2 \sigma_{\eta_i}^{-2} + (\sigma_{\lambda_i}^{-2}\mu_{\lambda_i} + \sigma_{\lambda_i}^{-2}\mu_{\lambda_i})^2(\sigma_{\eta_i}^{-2} + \sigma_{\lambda_i}^{-2} - 1)^{-1} \Big]
\end{aligned}
$$

where $\sigma_{\eta_i}^2, \sigma_{\lambda_i}^2, \mu_{\eta_i}, \mu_{\lambda_i}$ are the $i$th element of $\text{diag}^{-1}\Sigma_\eta$, $\text{diag}^{-1}\Sigma_\lambda$, $\mu_\eta$, $\mu_\lambda$, respectively. To effectively estimate $\sigma_{\lambda_i}^{-2} = \sum_{\mathbf{x} \in \mathbf{X}_n} \sigma_{h_i|f^{\times}}^{-2}$ and $\mu_{\lambda_i} = \sigma_{\lambda_i}^2 \sum_{\mathbf{x} \in \mathbf{X}_n} \sigma_{h_i|f^{\times}}^{-2}\mu_{h_i|f^{\times}}$, we can uniformly sample a mini-batch $\mathbf{X}_\mathcal{K}$ of size $K$ from $\mathbf{X}_\mathcal{D}$, and then compute the following noisy mini-batch

estimation:

$$\sigma_{\lambda_i}^{-2} = \sum_{\mathbf{x} \in \mathbf{X}_n} \sigma_{h_i|f\mathbf{x}_n}^{-2} = N \mathbb{E}_{\mathbf{x} \in \mathbf{X}_\mathcal{D}} \sigma_{h_i|f\mathbf{x}}^{-2} + \sum_{\mathbf{x}_l \in \mathbf{X}_n \setminus \mathbf{X}_\mathcal{D}} \sigma_{h_i|f\mathbf{x}_l}^{-2}$$

$$\approx \sum_{k \in \mathcal{K}} \frac{N}{K} \sigma_{h_i|f\mathbf{x}_k}^{-2} + \sum_{\mathbf{x}_l \in \mathbf{X}_n \setminus \mathbf{X}_\mathcal{D}} \sigma_{h_i|f\mathbf{x}_l}^{-2},$$

$$\mu_{\lambda_i} \sigma_{\lambda_i}^{-2} = \sum_{\mathbf{x} \in \mathbf{X}_n} \sigma_{h_i|f\mathbf{x}}^{-2} \mu_{h_i|f\mathbf{x}} = N \mathbb{E}_{\mathbf{x} \in \mathbf{X}_\mathcal{D}} \sigma_{h_i|f\mathbf{x}}^{-2} \mu_{h_i|f\mathbf{x}} + \sum_{\mathbf{x}_l \in \mathbf{X}_n \setminus \mathbf{X}_\mathcal{D}} \sigma_{h_i|f\mathbf{x}_l}^{-2} \mu_{h_i|f\mathbf{x}_l}$$

$$\approx \sum_{k \in \mathcal{K}} \frac{N}{K} \sigma_{h_i|f\mathbf{x}_k}^{-2} \mu_{h_i|f\mathbf{x}_k} + \sum_{\mathbf{x}_l \in \mathbf{X}_n \setminus \mathbf{X}_\mathcal{D}} \sigma_{h_i|f\mathbf{x}_l}^{-2} \mu_{h_i|f\mathbf{x}_l},$$

We denote the estimators for $\sigma_{\lambda_i}^{-2}$ and $\mu_{\lambda_i}$ by $\hat{\sigma}_{\lambda_i}^{-2}$ and $\hat{\mu}_{\lambda_i}$, respectively. Then, applying these noisy estimations to $\mathbb{E}_{n,\mathbf{X}_n \sim c} \mathbb{E}_{f \sim q_{\text{SPG}}(f)} \log \tilde{\mathcal{Z}}(\mathbf{f}^{\mathbf{X}_n})$, we have

$$\mathbb{E}_{n,\mathbf{X}_n \sim c} \mathbb{E}_{f \sim q_{\text{SPG}}(f)} \log \tilde{\mathcal{Z}}(\mathbf{f}^{\mathbf{X}_n})$$

$$\approx \frac{1}{2} \sum_{i=1}^{H} \mathbb{E}_{f \sim q_{\text{SPG}}(f)} \left[ \log \sigma_{\eta_i}^{-2} + \log \hat{\sigma}_{\lambda_i}^{-2} - \log(\sigma_{\eta_i}^{-2} + \hat{\sigma}_{\lambda_i}^{-2} - 1) \right.$$

$$\left. - \hat{\mu}_{\lambda_i}^2 \hat{\sigma}_{\lambda_i}^{-2} - \mu_{\eta_i}^2 \sigma_{\eta_i}^{-2} + (\hat{\sigma}_{\lambda_i}^{-2} \hat{\mu}_{\lambda_i} + \hat{\sigma}_{\lambda_i}^{-2} \hat{\mu}_{\lambda_i})^2 (\sigma_{\eta_i}^{-2} + \hat{\sigma}_{\lambda_i}^{-2} - 1)^{-1} \right], \mathbf{X}_n \sim c(\mathcal{T}^{\mathbb{Z}^+})$$

The symbol $\approx$ in the last line means that it is a consistent estimator, due to multivariate continuous mapping theorem. $\qquad\square$

### A.7 Proof of Proposition 7

**Proposition 7** (Debiasing). *Let $R$ be a random integer from a distribution $\mathbb{P}(N)$ that has support over the integers larger than $K$, and $\mathbf{x}_0$ is a random location sampled from $\mathcal{T}$. Then $\mathbb{E}_{n,\mathbf{X}_n \sim c} \mathbb{E}_{f \sim q_{\text{SPG}}(f)} \log \tilde{\mathcal{Z}}(\mathbf{f}^{\mathbf{X}_n})$ can be estimated by:*

$$\mathbb{E}\left[ \mathcal{J}_K + \sum_{k=K}^{R} \frac{\Delta_k}{\mathbb{P}(N \geq k)} \right] \tag{26}$$

*where $\Delta_k = \mathcal{J}_{\mathbf{k+1}} - \mathcal{J}_{\mathbf{k}}$, and the expectation $\mathbb{E}$ is taken over $R$, $n$, $\mathbf{X}_n$, and all mini-batches used by each $\mathcal{J}_k$ terms.*

**Proof** By definition, we have $\lim_{k \to \infty} \mathbb{E}\mathcal{J}_K = \mathbb{E}\mathcal{J}_N = \mathbb{E}_{n,\mathbf{X}_n \sim c} \mathbb{E}_{f \sim q_{\text{SPG}}(f)} \log \tilde{\mathcal{Z}}(\mathbf{f}^{\mathbf{X}_n})$. Apparently, $\mathbb{E} \sum_{k=0}^{R} \frac{\Delta_k}{\mathbb{P}(N \geq n)}$ constructs an Russian Roulette estimator [22]. Based on lemma 3 from [6], in order prove our result we only have to show that $\mathbb{E} \sum_{k=0}^{\infty} |\Delta_k| < \infty$. In fact, since the data distribution is assumed to be an empirical distribution, we have

$$\sum_{k=0}^{\infty} |\Delta_k| = \sum_{k=0}^{\infty} |\mathcal{J}_{\mathbf{k+1}} - \mathcal{J}_{\mathbf{k}}|$$

$$= \sum_{k=0}^{N-1} |\mathcal{J}_{\mathbf{k+1}} - \mathcal{J}_{\mathbf{k}}| < \infty$$

holds for all possible mini-batches used by each $\Delta_k$. The second equality is based on the fact that $\mathcal{J}_{\mathbf{k+1}} = \mathcal{J}_{\mathbf{k}} = \log \tilde{\mathcal{Z}}(\mathbf{f}^{\mathbf{X}_n})$ for all $k \geq N$. Therefore, we have $\mathbb{E} \sum_{k=0}^{\infty} |\Delta_k| = \sum_{k=0}^{N-1} \mathbb{E}|\Delta_k| < \infty$.

## B  Further details of experiments

### B.1  General settings

**Data split/Cross-validation schemes** For UCI experiments, each dataset was randomly split into train (90%) and test sets (10%). This was repeated 10 times. In contextual bandits, we used the code

open-sourced by [47], therefore the data sampling process described in [47] was exactly executed. For classification experiment for MNIST and CIFAR 10, since the train/test set are predefined, we have only run experiments with 5 different random seeds (for initialization). For interpolation with implicit prior experiment, see for details.

**Choice of sampling distribution 3** One example of $c$ that satisfies the requirement of Propositions 1, 2, and 3 takes the following form:

$$(n - |\mathcal{D}|) \sim \text{Geom}(p), \mathbf{x}_k \sim \mathcal{U}(\mathcal{T}), \ \ \forall 1 \leq k \leq n - |\mathcal{D}|,$$
$$\mathbf{X}_n := \mathbf{X}_\mathcal{D} \bigcup \{\mathbf{x}_k\}_{1 \leq k \leq n - |\mathcal{D}|} \tag{27}$$

where we first sample $n$ from a geometric distribution, such that $(n - |\mathcal{D}|) \sim \text{Geom}(p)$ with parameter $p$. Here, we use the parameter $p = 0.5$. Then, $(n - |\mathcal{D}|)$ out of distribution (OOD) measure points are sampled independently from a uniform distribution on $\mathcal{T}$.

**Choice of prior processes $p(f)$** Note that since FVI is an *inference method* instead of a new *model*, we will assume that FVI and most of the baselines will be using the same priors, whenever applicable. For example, in the interpolation structured prior tasks, both FVI and f-BNN will use the same piecewise implicit prior. In multivariate regression and image classification tasks, all algorithms will use the same BNN prior with the same structures, therefore we can isolate the difference caused by inference algorithms.

**Construction of the compact space $\mathcal{T}$** The construction of $\mathcal{T}$ depends on specific tasks. If we know the range of the input , we can directly set to be such interval. For example, in synthetic datasets of Experiment 6.1, we already know that the input lies in the interval between 0 and 1, therefore $\mathcal{T} = [0, 1]$. If we don't know the range of the inputs, then we can simply set $\mathcal{T}$ to be a hyperrectangle, with each $x_i \in [x^i_{min}, x^i_{max}]$, where $x^i_{min}$ and $x^i_{max}$ are the empirical min and max of the $i$-th variable of input dataset.

**Structure of SPGs** Unless specified otherwise, we use 10 basis functions for our SPGs, and each basis function is a two-layer neural network that maps from $\mathrm{R}^d$ to $\mathrm{R}^1$. The structure of these networks is input-100-100-output. Note that these neural network parameters are *not* part of the variational parameters, since they are frozen forever after we have finished distilling $p(f)$ using $\tilde{p}_{\text{SPG}}(f)$. To further reduce the number of free parameters, the parameters of the first two layers of all basis functions can be shared (this is applied only to larger scale experiments such as image classification). The encoder $\tilde{q}_\lambda(\mathbf{h}|f)$ for $\tilde{p}_{\text{SPG}}(f)$ is also a two-layer neural network (input-500-200-latent statistics), whose parameters are also fixed after distilling $p(f)$. The decoders also have two hidden layers (latent variables-50-100-output). The latent dimension is different depending on the tasks so that the comparison between baselines will be fair. This will be detailed later. For the stationary GP white noise process used in SPGs, we assume that they have isotropic noise level $\sigma_\nu^2 = 0.1$.

**Optimization** Unless otherwise specified, we use Adam optimizer with learning rate $lr = 0.001$. We use a slightly larger learning rate in the contextual Bandit experiment since the learning rates used for each baseline is tuned from $[0.001, 0.05]$, as specified in the experiment section. When training $\tilde{p}_{\text{SPG}}(f)$, we use 5k epochs unless otherwise noticed. For the inference phase where $q_{\text{SPG}}(f)$ is optimized to maximize the functional ELBO, the number of iterations is determined by the other baselines. For example in contextual Bandits, all baselines are trained for 100 epochs, so is FVI. In terms of batch size, unless specified otherwise, we choose the batch size to be 100. This batch size is also used to perform MC estimation of the likelihood term in Equation 20, and training $\tilde{p}_{SPG}$ in Equation 13.

**Hyperparameters of the likelihood function $p_\pi(y|f)$** Regarding the likelihood function $p_\pi(y|f)$, since we only deal with continuous outputs in this paper, we simply choose $p_\pi(y|f)$ to be a Gaussian noise with homogeneous standard deviation $\sigma$ and mean $\mu$ just like all the other baselines. The value of $\sigma$ is set to be 0.02 except for multivariate regression experiments, since we follow the setting of [28], where the noise variance is determined individually for all BNN baselines, including FVI. For FVI, we found that making the mean function $\mu$ of $p_\pi(y|f)$ to be optimizable will accelerate the convergence (which is equivalent to adding an additional basis/bias in SPGs).

**Optimize prior or not**   Sometimes, it would be interesting to see how inference methods would perform given their respectively "best" prior. That is, we may further optimize the parameters of prior, $p(f)$. However, our paper mainly treats FVI/f-BNN/VIPs/MFVIs, etc as *inference methods* rather than *standalone models*. From this perspective, we are more interested in the case where the prior is fixed (hence the ground truth posterior is given). That being said, if we were to treat the approximate predictive distribution $q(f)$ as a standalone model, then indeed it makes sense to seek the optimal prior for each method. We leave this topic for more future works.

**Computational resources**   Synthetic experiment/contextual bandits are mostly run on a laptop using only the CPUs. Other experiments are mostly done on a machine with NVIDIA Quadro P5000 GPU plus 30GB of memory.

## B.2   Individual settings for each experiments

**Interpolation with structured implicit priors**   To sample a random function from piecewise constant priors, we first sample the number of change points $n$ from a Poisson distribution $\mathrm{Poisson}(3)$. The exact location of each change point is uniformly sampled from $[0, 1]$. Then, for each interval specified by the change points, we sample $n + 1$ function values uniformly to specify constant function values in each interval. This results in piecewise constant functions. For the piecewise linear prior, the function values are sampled similarly and we draw straight lines to connect each function value. In this experiment, for FVI, we draw 1k samples from the implicit priors that are used to optimize the FVI parameters. The basis function for FVI are directly obtained by sampling from the implicit prior. We use 200 basis functions, with a latent dimension of 10. To train our $\tilde{p}_{\mathrm{SPG}}(f)$, we sample 1k random function samples and optimize $\tilde{p}_{\mathrm{SPG}}(f)$ for 5k epochs. For inference, the variational parameters are trained for 1k epochs.

**UCI Multivariate regression**   For this experiment, we follow [28]. The functional prior is a fully connected ReLU BNN with two hidden layers (input-10-10-output). We train FVI for 1k epochs. We use 10 basis functions for FVI and a batch size of 100. The latent size is set to 100.

**Gaussian Processes in UCI regression**   On UCI datasets, variational sparse GPs and exact GPs are implemented using GPflow. VSGPs uses 50 inducing points. Both variations of GP models use the RBF kernel.

**VIPs, Bayes-by-Backprop, variational dropout, $\alpha$-dropout for Bayesian neural networks on UCI**   VIP and Weight-space inference methods for BNNs are based on reimplementations provided by [28]. For details of othese, readers may refer to Appendix F.3 of [28]. Results in Table 1 are taken from [28], as they have used exactly the same data split scheme and BNN prior structures.

**Contextual Bandits**   We use similar settings to [47], where we use a batchsize = 32, training epochs = 100, training frequency = 50, and contexual points = 2000. We use ReLU BNNs as functional priors. It has two hidden layers, each with 50 hidden units. For FVI, we use 100 basis functions (with shared weights until the last layer), a learning rate of 0.005. For details of the algorithms mentioned in Table 2, readers may refer to [37] for details. Here we briefly explain the meaning of the abbreviations. **FVI**: functional variational inference; **FBNN**: functional Bayesian neural networks; **Uniform**: uniform sampling; **RMS**: trains a neural network and acts greedily using RMSprop; **Boot RMS**: Bootsrapped RMS; **Neural Linear**: Bayesian linear regression over deep NN features; **ParamNoise**: just a regular DNN, but when making decisions, an isotropic Gaussian perturbation is added to the NN weights; **Dropout**: variational dropout BNNs; **BBB**: Bayes-by-Backprop BNNs; **BB $\alpha$**: Black-box alpha divergence minimization.

**Image classification and OOD detection**   For all models in this experiment, the CNN structures are the same as in [21]. That is, the 3 convolutional layers plus 3 fully connected layers in the DeepOBS benchmark [43]. Similar to [21], we apply standard isotropic Gaussian prior on all weight parameters. We use Adam with learning rate of 0.001 and batch size of 100, and run the training procedure for 100 epochs. For FVI, we use 100 basis functions in the SPGs on both datasets. Note that each basis function is a there-layer convolutional network that maps from $\mathrm{R}^d$ to $\mathrm{R}^{10}$. To significantly reduce the memory usage, the parameters of the convolutional layers of all basis functions are shared.

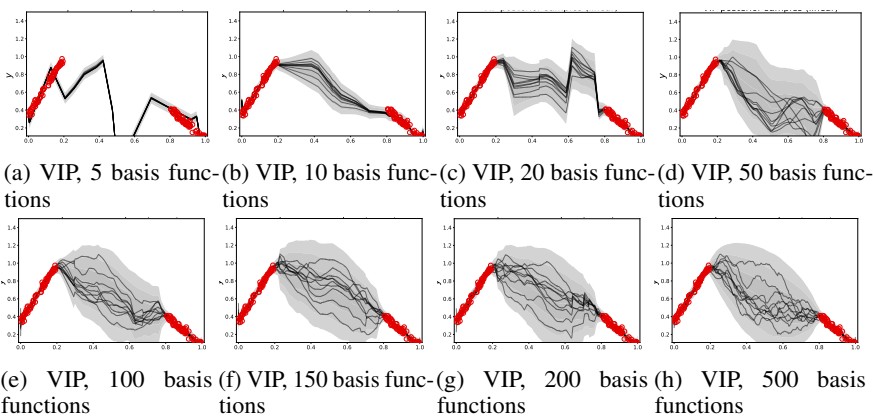

(a) VIP, 5 basis func-(b) VIP, 10 basis func-(c) VIP, 20 basis func-(d) VIP, 50 basis func-
tions                  tions                  tions                  tions

(e)  VIP,  100  basis (f) VIP, 150 basis func-(g)  VIP,  200  basis (h)  VIP,  500  basis
functions             tions                  functions             functions

Figure 4: The posterior samples from VIPs with different number of basis functions. As more
basis functions are used, the posterior samples from VIP become more and more noisy, and finally
converges to GP-like behaviour when 500 basis functions are used. Compared to the ground truth
estimate from Figure 3 in the paper, VIP clearly under-estimates the predictive uncertainties in-
between the training samples.

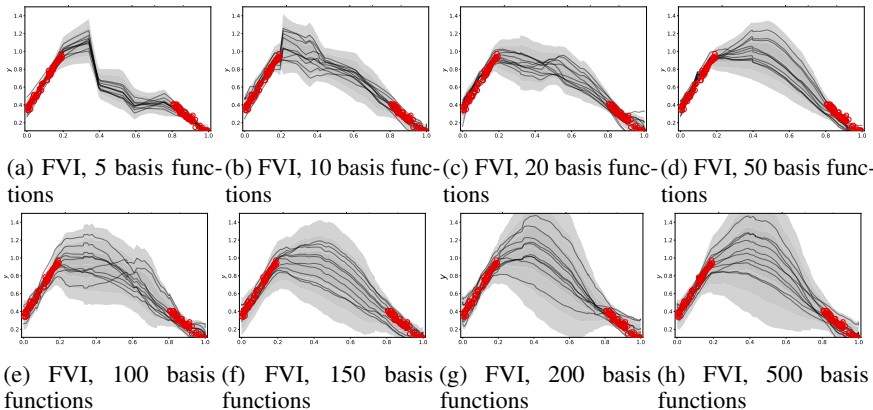

(a) FVI, 5 basis func-(b) FVI, 10 basis func-(c) FVI, 20 basis func-(d) FVI, 50 basis func-
tions                  tions                  tions                  tions

(e)  FVI,  100  basis (f)  FVI,  150  basis (g)  FVI,  200  basis (h)  FVI,  500  basis
functions             functions             functions             functions

Figure 5: The posterior samples from FVI with different number of basis functions. FVI is still
able to learn the piecewise linear behaviour from the prior as more basis functions are used. As the
number of basis functions is increased to 500, FVI converges to a solution that is much closer to the
ground truth (compared with VIP), and is still able to exhibit non-Gaussian behaviours from the prior.

## C   Additional Experiments

### C.1   Impact of number of basis functions on SPGs and VIPs

As discussed in Section 4.2, the SPGs used the proposed FVI can be treated as a non-Gaussian
extension of the VIP variational family, by removing the Gaussian assumption on $\mathbf{a}$. One natural
question that arises in this setting will be that, does the advantage of FVI over VIP vanish as the
number of basis function increases? How does the number of basis functions affect the performance
of each method? To provide more intuition for FVI and VIPs, we consider again the 1-D function
interpolation task with piecewise-linear implicit prior. In Figure 4 and Figure 5, we show how the
posterior samples from FVI and VIPs evolve when the number of basis functions gradually increase
from 5 to 500. Note that for a fair comparison, the basis functions for both FVI and VIP are obtained
by drawing random function samples from the implicit prior. Since the piecewise-linear implicit
functional prior is not reparameterizable, once the basis functions for FVI and VIP are sampled,
they will be frozen forever (in contrast, when the prior is reparameterizable, both FVI and VIP can
optimize the basis functions, therefore the number of basis functions required will be much smaller
than this experiment).

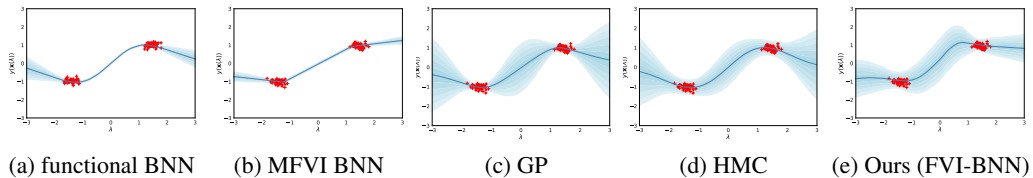

| (a) functional BNN | (b) MFVI BNN | (c) GP | (d) HMC | (e) Ours (FVI-BNN) |

Figure 6: A regression task on a synthetic dataset (red crosses) reproduced from [10]. We plot predictive mean and uncertainties for each algorithms. This tasks is used to demonstrate the theoretical finding on the pathologies of weight-space VI for single-layer BNNs: there is *no* setting of the variational parameters that can model the in-between uncertainty between two data clusters. The functional BNNs [47] also has this problem, since BNNs are use as part of the model. On the contrary, our functional VI method can produce sensible in-between uncertainties for out-of-distribution data. See Appendix C.2 for more details.

From Figure 4 and Figure 5, we can first observe that as the number of basis function increases, the predictive uncertainty of both FVI and VIP also increase, until around when 200 basis functions are reached. However, as more basis functions are used, the posterior samples from VIP become noisier, and finally converges to GP-like behavior when 500 basis functions are used. Compared to the ground truth estimate from Figure 3 in the paper, VIP under-estimates the predictive uncertainties in-between the training samples. This is due to that the piecewise linear behavior of the function samples violates the Gaussian assumption of VIP, such that the correlation level between points will be lower than expected. On the other hand, FVI is still able to learn the piecewise linear behavior from the prior as more basis functions are used. As the number of basis functions is increased to 500, FVI converges to a solution that is much closer to the ground truth (compared with VIP) and is still able to exhibit non-Gaussian behaviors from the prior. We can conclude that the advantage of FVI over VIP does not vanish as the number of basis functions increases. In contrast, the difference between FVI and VIP becomes even more distinct and recognizable.

## C.2   On in-between uncertainty pathologies of BNNs

In figure 6, we presented a 2-D regression tasks on a synthetic dataset (red crosses), reproduced from [10]. This tasks is used to demonstrate the pathologies of weight-space inference for single-layer BNNs (including f-BNNs where BNNs are use as part of the model): there is *no* setting of the variational parameters that can model the in-between uncertainty between two data clusters. To be concrete, we have the following proposition:

**Proposition 8** (Limitations for single-hidden layer BNNs [10]). *Consider any single-hidden layer fully-connected ReLU NN* $f : \mathbb{R}^D \to \mathbb{R}$. *Let* $x_d$ *denote the* $d^{th}$ *element of the input vector* **x**. *Suppose we have a fully factorised Gaussian distribution over the weights and biases in the network. Consider any points* $\mathbf{p}, \mathbf{q}, \mathbf{r} \in \mathbb{R}^D$ *such that* $\mathbf{r} \in \overrightarrow{\mathbf{pq}}$ *and either:*

    *i. $\overrightarrow{\mathbf{pq}}$ contains $\mathbf{0}$ and $\mathbf{r}$ is closer to $\mathbf{0}$ than both $\mathbf{p}$ and $\mathbf{q}$.*

    *ii. $\overrightarrow{\mathbf{pq}}$ is orthogonal to and intersects the plane $x_d = 0$, and $\mathbf{r}$ is closer to the plane $x_d = 0$ than both $\mathbf{p}$ and $\mathbf{q}$.*

*Then* $\mathrm{Var}[f(\mathbf{r})] \leq \mathrm{Var}[f(\mathbf{p})] + \mathrm{Var}[f(\mathbf{q})]$.

That is, the weight space inference of a single-hidden layer variational BNNs (using mean-field VI) fails to represent the in-between uncertainty, and become over-confident on out-of-distribution data. In this experiment, the training data is sampled as follows: the 2-D input locations of training data are generated by sampling 100 points, 50 each from two separate clusters that follow Gaussian distributions. The inputs of the cluster on the left of Figure 1 around $(-1, -1)$, and the other cluster is centered around $(1, 1)$. Both have isotropic Gaussian noise with zero mean and variance 0.01. The outputs ($y$) are -1 and 1 for the left and right clusters, respectively. We further add a Gaussian observational noise of variance 0.1 to the outputs. To test whether the baselines can learn the in-between uncertainties between clusters, we use a fully connected ReLU BNN of a single hidden layer (50 units). The FVI also uses this prior as functional prior and has 50 basis functions and 50 latent dimensions. The settings of MFVI are determined according to [10]. The settings of F-BNN are determined similarly.

In figure 1, the $\lambda$ axis is the 1-D parameter that parameterizes the 1-dimensional straight line embedded in the 2-D plane, that connects $(-3, -3)$ and $(3, 3)$. The value of the $\lambda$-coordinate implies that its actual 2-D coordinate in the 2-D plane is $(\lambda, \lambda)$. The results in Figure 1 show that both Mean-field variational BNN and functional BNN suffers from the limitations of single hidden layer BNN. On the contrary, FVI can produce a sensible in-between uncertainty that is similar to GPs and HMC. For GPs, we use infinite-width BNN kernel following [10]

### C.3 CPU time comparison, FVI vs f-BNN on implicit priors

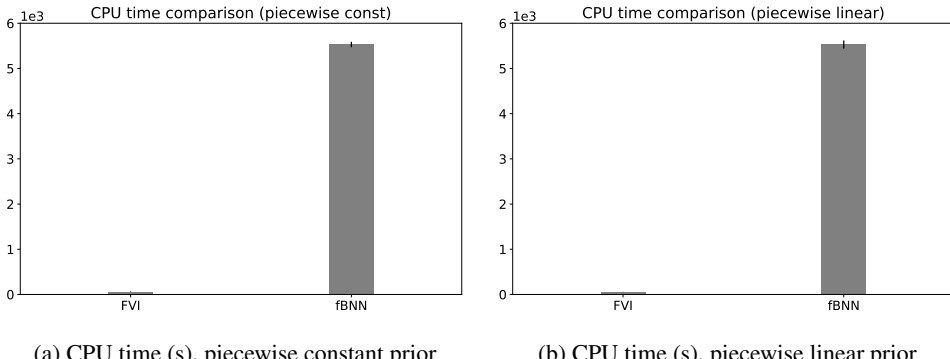

(a) CPU time (s), piecewise constant prior   (b) CPU time (s), piecewise linear prior

Figure 7: CPU time comparison, FVI vs f-BNN on implicit priors. Although f-BNNs are only trained for 100 epochs, its running time is still 100x slower than FVI.

### C.4 CPU time comparison, FVI vs f-BNN on Census

In order to compare the efficiency between FVI and f-BNN, we provide the CPU time comparison of running contextual bandits on Census dataset [1], one of the largest contextual bandits dataset that we have tested. It has more than 2 million data points, each with 389 dimensional features as input (including dummy binary variables for categorical variables). The output has 9 different classes (actions). The CPU time consumed by each algorithm on Census is listed as follows:

Table 4: CPU time performance comparison of running contextual bandit on Census dataset

|  | FVI | f-BNN | BBB |
|---|---|---|---|
| Run time (s) | $28.14 \pm 2.604$ | $9648.19 \pm 957.3$ | $19.98 \pm 1.238$ |

Based on Table 4, we can see that FVI is nearly 500 times faster than f-BNN. The run time of FVI is similar to Bayes-by-Backprop, indicating that FVI is very efficient and scalable.

### C.5 Improved results of f-BNN on implicit priors

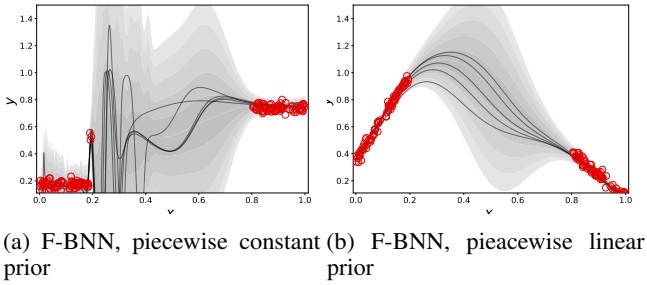

(a) F-BNN, piecewise constant prior   (b) F-BNN, pieacewise linear prior

Figure 8: F-BNN on structured implicit priors, trained with 10k epochs

In experiment 6.1, we have only run f-BNNs for 100 epochs due to its computational costs. Here, we provide improved results of fully-trained f-BNNs after 10k epochs. Note that this epoch number

is much larger than the FVI setting (5k), since we found that after 5k epochs, the f-BNN posteriors do not seem to improve over the results in experiment 6.1. As shown in Figure 8, after 10k epochs, the posterior uncertainty estimates of f-BNNs become much loser to the ground truth in Figure compared with its 100 epochs version. However, this comes with the cost of significantly increased computational time. Moreover, f-BNNs seem to provide less convincing posterior samples in terms of mimicking the piece-wise constant/linear behaviours of the implicit priors.

## C.6 Large scale experiments on deeper Bayesian neural networks

To demonstrated the scalability and applicability of FVIs to larger datasets and deeper Bayesian neural networks, in this section we perform regression experiments using a Bayesian DNN with 5 hidden layers of 100 units. We compare our results to f-BNNs and BBB with the same DNN structure, which are cited directly from [47]. For fair comparison, we increase the size of the basis functions used by SPGs to neural networks with 5 hidden units and 100 units. An additional hidden layer is added to the decoder and encoder of the VAE used by SPGs. We follow the settings of Section 6.2, except that we train FVI using 30000 iterations using mini-batch stochastic optimization. We report results on Naval dataset, protein datset, and GPU dataset. We also include results of FVI on shallow networks used in Section 6.2. From Table 6, we notice that the performance of FVI on deeper networks is generally competitive to f-BNNs and BBBs, indicating that FVI is scalable to larger datasets and deeper neural networks.

## C.7 Comparison to function space particle optimization (f-SVGD) and GPs

In this section, we further compare FVI to function space particle optimization (f-SVGD) and GPs:

Table 5: Regression experiment: Average test negative log likelihood

| Dataset | N | FVI | f-SVGD | GP |
|---------|---|-----|--------|-----|
| boston | 506 | 2.33±0.04 | **2.30±0.05** | 2.63±0.04 |
| concrete | 1030 | **2.88±0.06** | 2.90±0.02 | 3.4±0.01 |
| energy | 768 | **0.58±0.05** | 0.69±0.03 | 2.31±0.02 |
| kin8nm | 8192 | **-1.15±0.01** | -1.11±0.01 | -0.76±0.00 |
| power | 95684 | **2.69±0.00** | 2.73±0.00 | 2.82±0.00 |
| protein | 45730 | **2.85±0.00** | 2.85±0.00 | 3.01±0.00 |
| red wine | 1588 | 0.97±0.06 | **0.89±0.01** | 0.98±0.02 |
| yacht | 308 | **0.59±0.11** | 0.75±0.01 | 2.29±0.03 |
| naval | 11934 | -7.21±0.06 | -4.82±0.10 | **-7.81±0.00** |

Note that f-SVGD is not included in our main experiments in Table 1, since it is a particle optimization-based inference method. On the other hand, GP is not included since it is not a BNN-based model. For GPs, we used variational sparse GP with 50 inducing points plus an RBF kernel. The additional results in Table 5 shows that FVI performs the best in 6 out of 9 datasets. Moreover, FVI outperforms f-SVGD in 6 out of 9 datasets and outperforms GP in 8 out of 9 datasets in terms of NLLs.

Table 6: larger scale regression experiment: Average test negative log likelihood

| Dataset | N | FVI | f-BNNs | BBB | FVI shallow |
|---------|---|-----|--------|-----|-------------|
| GPU | 241600 | **2.93±0.03** | 2.97±0.02 | 2.99±0.01 | 3.10±0.04 |
| Protein | 45730 | 2.82±0.01 | 2.72±0.01 | **2.72±0.01** | 2.85±0.00 |
| Naval | 11934 | **-7.42±0.01** | -7.24±0.01 | -6.96±0.01 | -7.38±0.04 |

### C.8 Out-of-distribution detection visualization on CIFAR10

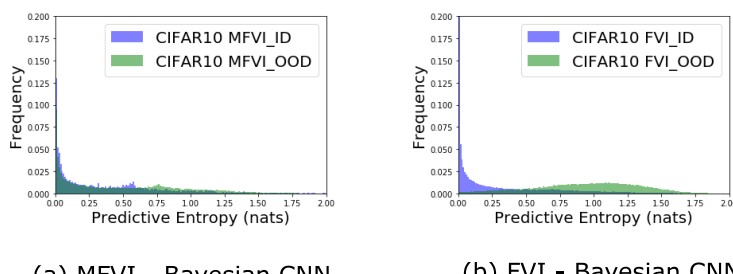

(a) MFVI - Bayesian CNN      (b) FVI - Bayesian CNN

Figure 9: Histograms of predictive entropies on CIFAR10/SVHN OOD detection. Left: MFVI. Right: FVI.