# OpenReview forum: "Functional Variational Inference based on Stochastic Process Generators"
_NeurIPS.cc/2021/Conference — NeurIPS 2021 Poster_

### Official Review · Reviewer_YNdn · 2021-07-01

**Rating:** 7
**Confidence:** 2

**Summary:**

The paper presents an approach to probabilistic regression. The posterior distribution is estimated using a variational approach. A variational class of stochastic processes is presented, which is parameterized by neural networks - amongst other things by a product of experts variational autoencoder. For optimizing the variational distribution a variational bound and its sample approximation is derived. The proposed approach is evaluated on regression, classification and reinforcement learning tasks.

**Main Review:**

Overall, I enjoyed reading the paper and the results seem to be very competitive and impressive.

Originality: The approach seems like a new approach, which combines and extends ideas from previous work, such as [44], [27]  and [50]. The techniques are as far as I could tell well known. The paper differentiates well from the related work, which is sufficiently discussed.

Quality: As far as I could tell the submission is technically sound. The resulting algorithm is extensively evaluated empirically and some theoretical inside are given.

Clarity: The paper is nicely written and organized. I think that it adequately informs the reader.

Significance: The empirical results seem pretty impressive and could be of much use to the machine learning community.


One thing I missed was:
- An experiment where the true posterior is a Gaussian process and the proposed approach is evaluated against it. In such an example the ground truth is not only given by samples from the posterior as in Sec. 5.1, but the whole posterior distribution can be evaluated in closed form. As such, one could evaluate how accurate the variational approach is.

Things I did not understand:
- Why is in eq. (12) the prior only fitted using regressors $\mathbf X_O \subseteq \mathbf X_{\mathcal{D}}$ from the data and not some arbitrary regressors $\mathbf X_O \subseteq \mathcal T^{\mathbb Z_+}$?
- Why is FBNN performing so poorly in Figure 2? The example seems extremely simple and Figure 3 in [44] suggest different results. If it is just because of the training time needed, I do not that such a comparison is fair.

Remarks:

-Reference [44] is missing the venue.

**Time Spent Reviewing:**

6

---

> ### Author Response · Authors · 2021-08-10
> **Response to Reviewer YNdn**
>
> Thank you very much for your review. We are pleased to see that you appreciate the effort and novelty of our work.
>
> > **Q1: An experiment where the true posterior is a Gaussian process and the proposed approach is evaluated against it. In such an example the ground truth is not only given by samples from the posterior as in Sec. 5.1 but the whole posterior distribution can be evaluated in closed form. As such, one could evaluate how accurate the variational approach is.**
>
> **A1:** Thanks for the suggestion. In Appendix D.2, we actually considered a very similar experiment setting: 1D regression using BNN priors. In this case, we expect that the true posterior would be close to a GP (due to BNN-GP correspondence), and both the HMC estimation and GP/NTK posterior are tractable. As such, we could investigate how good these different variational inference methods are. In revision, we will try to add more examples where the ground truth posterior is analytically tractable.
>
> Additionally, we would like to point out that the main focus of our method is to solve the inference problem of non-GP priors. Therefore, we constructed the synthetic example in Section 5.1, and evaluate the ground truth posterior using MC samples. We hope these experiments will complement each other.
>
> > **Q2: Why is in eq. (12) the prior only fitted using regressors $\mathbf{X}\_O \subset \mathbf{X}\_{\mathcal{D}}$
>  from the data and not some arbitrary regressors $\mathbf{X}\_O \subset \mathcal{T}^{\mathrm{Z}^+}$
> ?**
>
> **A2:** From an algorithmic point of view, both options are valid, and using $\mathbf{X}\_O \subset \mathcal{T}^{\mathrm{Z}^+}$ might actually be a good idea depending on the scenarios. We chose to use $\mathbf{X}\_O \subset \mathbf{X}\_{\mathcal{D}}$ mostly due to its practical merit: for stochastic processes with high dimensional inputs (especially those without a low dimensional latent structure), it might not be efficient to fit the prior on arbitrary regressors $\mathbf{X}\_O \subset \mathcal{T}^{\mathrm{Z}^+}$. Instead, we might want to focus on only *important* points (i.e., the training set $\mathbf{X}\_O \subset \mathbf{X}\_{\mathcal{D}}$), on which we would like to match our SPG to the prior.
>
> > **Q3: Why is FBNN performing so poorly in Figure 2? The example seems extremely simple and Figure 3 in [44] suggest different results. If it is just because of the training time needed, I do not that such a comparison is fair.**
>
> **A3:** Firstly, as mentioned in Section 5.1, f-BNNs are trained for 100 epochs, since their running time is higher than that of FVI. We agree that if f-BNNs are only *marginally* slower than other methods, then this might not be a fair comparison. However, it is not the case. As shown in Appendix D.3, the running time of f-BNNs is hundreds of times slower than other methods, even with smaller epochs. At last, after the NeurIPS submission, we have also obtained the f-BNN posterior samples after convergence, which is still worse than our method. We will add the converged sample plots in revision.

---

> > ### Comment · Reviewer_YNdn · 2021-08-23
> > **My recommendation remains the same**
> >
> > Thank you for your answers. My concerns are addressed. My recommendation remains the same.

---

### Official Review · Reviewer_Jd3R · 2021-07-02

**Rating:** 6
**Confidence:** 3

**Summary:**

The authors propose "Functional Variational Inference" (FVI), a novel variational approach to approximate Bayesian inference in function space. The approach is based on the "grid-functional KL divergence", a novel form of KL divergence between stochastic processes. The authors show that this is a proper KL-divergence and that it is well-defined in cases where previous work (fBNN) is not.

Furthermore, the authors propose "Stochastic Process Generators" (SPGs), a novel family of variational distributions which are more flexible in comparison to distributions used in previous work (VIP). Using SPGs, the authors derive an efficient approximation of the ELBO corresponding to the grid-functional KL divergence. The authors demonstrate that this approximation scales better than the one used in fBNN.

The authors evaluate FVI on a range of experiments (interpolation with implicit priors, regression and classification with BNN priors, Thompson sampling). FVI tends to perform favourably in comparison to fBNN, VIP, and a range of other existing methods.

**Ethical Concerns:**

No ethical concerns.

**Limitations And Societal Impact:**

No potential negative societal impact.

**Main Review:**

Originality: The proposed FVI is a novel variational approach in function space. The authors explain in which respect their work extends and improves upon prior approaches (fBNN and VIP). As described below (Sec. "Clarity"), the paper could be strengthened by providing a bit more intuition on how FVI remedies the weaknesses of prior work.

Quality: The proposed method appears to be formulated in a technical sound way (I did not check all of the proofs). The authors establish the well-definedness of FVI and provide both theoretical and empirical evidence that their method improves upon prior work.

Nevertheless, there are some weaknesses, on which I ask the authors to elaborate:

- The authors use the Product of Experts (PoE) architecture for the encoder used to distill the prior into an SPG. While I think this is a valid approach, I do not understand the author's comments on the "Deep Set"-like approach used for the Neural Process (NP) family. Why should this method be inferior to the PoE approach, i.p., why should it be more "difficult in high-dimensional spaces" (l. 219). Also, what do the authors mean with "need to model all possible conditionals" (l. 222). Did the authors try the "Deep Set"-like approach for their method?
- The results for fBNN in Fig. 2 seem to be way worse than those shown by the original authors on a similar experiment, cf. Sun et al., "Functional Variational Bayesian Neural Networks" Fig. 3. Could the authors elaborate on this discrepancy? What do the authors mean with "for fairness" in l. 247?
- Why do the authors consider an additive white noise process $\nu$ in their definition of the SPG? Isn't noise already considered through the likelihood $p_\pi$?

Clarity: Overall, the writing style and paper organisation is good. Nevertheless, I think some points could be improved. As FVI is closely related to fBNN and VIP, I would like to see a bit more background on those methods and how FVI remedies their shortcomings. In particular, it would help the reader if the authors could provide some intuition for the following (I would suggest moving some of the more technical details to the appendix):

- What is the intuition behind using Eq. (3) instead of Eq. (2)? Why is it crucial to have a random number $n$ of measurement points in Eq. (3)? What is the influence of the sampling distribution $c$ in Eq. (3) and of the particular choice Eq. (6)? I am aware that the authors provide formal arguments to support these choices, but I think that their presentation lacks intuitive motivation for them.
- Why does it make sense to freeze the basis functions and decoder parameters after fitting the prior using an SPG? Is this merely a choice required to make the KL divergence between SPGs tractable? How does this choice relate to Proposition 4, i.e., how does this "parameter freezing" affect SPG's expressiveness?

Minor concerns/typos (do not influence my score):

- l. 26: missing space
- l. 28: F-BNN → f-BNN
- l. 56: to generated → to generate
- l. 68: supreme → supremum
- l. 88: exist → exists
- l. 97: $X_\mathcal D$ is undefined
- Eq. (8): the integral is missing a closing $\mathrm d h$. The same goes for all other appearances of this VAE integral.
- l. 126: Then, for $\forall$ → Then, $\forall$
- Text around Eq. (11) states twice that Eq. (11) is called $\tilde p_{SPG}$.
- ll. 140-142: $f_m^X$ is explained twice
- l. 157: extra comma
- Eq. (19): The likelihood misses a subscript $\pi$

Significance: The authors propose an interesting approach to an important problem. The results show improvements compared to previous work. Further, the method seems to be theoretically well-founded. I therefore think the proposed FVI can be a valuable contribution. However, in its current form, I judge the paper to be borderline because of the concerns I raised above. If the authors could elaborate and improve upon those points, I am willing to raise my score.

------------------------------------------------------------------------------
Update after reading the author's response:

I thank the authors for their response. I raise my score because the authors addressed my concerns and will clarify some of the points I raised in the updated manuscript.

**Time Spent Reviewing:**

5

---

> ### Author Response · Authors · 2021-08-10
> **Response to Reviewer Jd3R**
>
> We thank the reviewer for his/her valuable time for detailed reviews on our submission. Here are our responses to the concerns raised by the reviewer.
>
> > **Q1: The authors use the Product of Experts (PoE) architecture for the encoder used to distill the prior into an SPG. While I think this is a valid approach, I do not understand the author's comments on the "Deep Set"-like approach used for the Neural Process (NP) family. Why should this method be inferior to the PoE approach, i.p., why should it be more "difficult in high-dimensional spaces" (l. 219).**
>
> **A1:** Sorry for the confusion, we will make relevant discussions in our paper more clear in revision. We did not mean to imply that deep-set is inferior to our PoE. In fact, our PoE approach *is* a particular way of parameterizing a Deep Set encoder (in the context of VI), therefore they can be used interchangeably. What we wanted to convey in lines 219-220 is: *the way that we utilize* the deep-set/PoE encoders is what makes our approach more efficient. Let us elaborate this a little bit more below.
>
> * First, when we say the deep-set approach is difficult "in high-dimensional spaces'', we were referring to the dimensionality of *rows* (number of data points of $\mathbf{X}_O$) rather than columns (number of features of each data point $\mathbf{x} \in \mathbf{X}_O$ ).
>
> * Imagine that we want to calculate $\tilde{q}_\lambda(\mathbf{h}|\mathcal{D})$ conditioned on millions of data points $\mathbf{X}_O$, it is not very efficient to feed *all* data points to the deep set/PoE encoder.  This argument is true for *both* PoE and general Deep Sets (as commented in line 178).
>
> * The advantage of our approach lies in: i) during training time, we can construct a mini-batch estimator (Proposition 6), thus avoiding the evaluation of the inference net $\tilde{q}\_\lambda$ on a (possibly) huge collection of data points; ii), during test time, when computing predictive distribution $q_{SPG}(f)$, we only need $q\_\eta(\mathbf{h})$, which is just a simple factorized Gaussian distribution (no PoE/Deep sets involved).
>
> * On the contrary, in neural processes, during test time, they still need to pass the training set ("context points'') $\mathcal{D}$ to the inference net $q(\mathbf{z}|\cdot)$, and then generate a prediction via $p(y^*|\mathbf{z},\mathbf{x}^*)$.
>
> > **Q2: The results for fBNN in Fig. 2 seem to be way worse than those shown by the original authors on a similar experiment, cf. Sun et al., "Functional Variational Bayesian Neural Networks" Fig. 3. Could the authors elaborate on this discrepancy? What do the authors mean with "for fairness" in l. 247?**
>
> **A2:** As mentioned in Section 5.1 and Appendix D.3, the running time of f-BNNs are hundreds of times slower than other methods. Therefore, for fairness in terms of computation time, f-BNNs are trained for 100 epochs (instead of 20k in the original f-BNN paper). In fact, after the NeurIPS submission, we have also obtained the f-BNN posterior samples after convergence, which is still worse than our method. We will add the converged sample plots in revision.
>
> > **Q3: Why do the authors consider an additive white noise process $\nu$ in their definition of the SPG? Isn't noise already considered through the likelihood $p_\pi$ ?**
>
> **A3:**  First, note that $p_\pi$ is not always a Gaussian likelihood; for example, in classification problems, $p_\pi$ can be a class-conditional distribution. In this case, noise is not considered through $p_\pi$. Second, the reason that we added $\nu$ to SPGs is also practical: we need to use SPGs to distill/approximate the prior, $p(f)$, by optimizing Equation (12). By adding a white noise $\nu$, we are essentially introducing a Gaussian likelihood function (i.e., $\tilde{p}(\mathbf{f}_m^{\mathbf{X}_O}|\mathbf{a})$), which allows us to perform unsupervised learning by optimizing Eq. (12).
>
> > **Q4: Overall, the writing style and paper organisation is good. Nevertheless, I think some points could be improved...**
>
> **A4:**  Thank you so much for your suggestion regarding improving the clarity. We will improve the presentation of the paper, and provide more intuition regarding the objective (Equation (3)) and the parameter freezing procedure for SPGs.
>
> ---
> ---
> (*Responses below are included for completeness.*)
>
> > **Q5: What is the intuition behind using Eq. (3) instead of Eq. (2)? Why is it crucial to have a random number $n$ of measurement points in Eq. (3)? What is the influence of the sampling distribution $c$ in Eq. (3) and of the particular choice Eq. (6)? I am aware that the authors provide formal arguments to support these choices, but I think that their presentation lacks intuitive motivation for them.**
>
> **A5:** Indeed, most of our supports to those choices are quite formal. This is because Eq (3) is trying to address a formal question: can we propose a new functional divergence, such that it is more well-defined than (2) (i.e., it is finite under weaker assumptions)? Our proposed objective (Eq (3)) is defined by the expectation/integration operator (on the random number $n$ of measurement points). Intuitively, this makes sense, since the expectation/integration operator should be more computationally robust than the maximization/supremum operator (used in Eq (2)).
>
> > **Q6: Why does it make sense to freeze the basis functions and decoder parameters after fitting the prior using an SPG? How does this choice relate to Proposition 4, i.e., how does this "parameter freezing" affect SPG's expressiveness?**
>
> **A6:** Theoretically speaking, the expressiveness result of Proposition 4 does not require us to optimize the basis functions, thus it does not prevent us to stick to a set of fixed basis functions (either predefined or learned from data). In this paper, we chose to fix basis functions after fitting the prior, since 1) it's convenient and already gives good enough results, and 2), it can keep the KL divergence in Eq (15) tractable.
>
> For completeness, we would like to point out that after fitting the prior using $p_{SPG}$, it is still possible to *free* the parameters of a *subset* of the basis functions for $q\_{SPG}(f)$. We may continue to optimize these suset of basis functions, while keeping the KL divergence between two SPGs tractable. For example, the SPG prior $p_{SPG}(f)$ could be defined as $f = \sum_s a\_s \phi\_s(\cdot,\mathbf{w}\_s) + \sum\_r b\_r \varphi\_r(\cdot, \mathbf{\mathbf{m}}\_r) + \nu$. Here, $\\{ \phi\_s \\}$ is the set of basis function used in the paper (will be fixed when optimizing ELBO in Eq (17), as in the paper). $\\{\varphi\_r\\}$ is the set of basis functions that will *not* be fixed when optimizing Eq (17), and $\\{\mathbf{\mathbf{m}}\_r\\}$ are their parameters. $\mathbf{a} \sim \int\_{\mathbf{h}} p\_\theta(\mathbf{a}|\mathbf{h}) p\_{0}(\mathbf{h})$ is the non-Gaussian coefficients generated by a VAE decoder, and $\mathbf{b} \sim \mathcal{N}(\mathbf{b}; 0,\xi^2)$ is the zero mean Gaussian weights with *small enough, fixed* variance $\xi$.  Then, the variational distribution $q_{SPG}$ could be defined as $f = \sum\_s a\_s \phi\_s(\cdot,\mathbf{w}\_s) + \sum\_r b\_r \varphi\_r(\cdot, \mathbf{\mathbf{m}}\_r) + \nu$, where $\mathbf{a} \sim \int\_{\mathbf{h}} p_\theta(\mathbf{a}|\mathbf{h}) q_{\eta}(\mathbf{h})$ and $b\_r \sim \mathcal{N}(b\_r; \alpha\_r,\beta^2\_r)$, here, $q\_{\eta}(\mathbf{h})$, $\\{\mathbf{\mathbf{m}}\_r\\}$, and
> $\\{\alpha\_r, \beta^2\_r\\}$ are all optimizable when maximizing the grid-functional ELBO in Equation (17). In this way, the (grid-functional) KL divergence between $q\_{SPG}$ and $p\_{SPG}$ is still tractable, similar to Eq (15).

---

### Official Review · Reviewer_4izH · 2021-07-13

**Rating:** 7
**Confidence:** 4

**Summary:**

This paper worked on the approximation for the posterior distribution, especially when we perform Bayesian inference in the space of functions. The authors provided a new Kullback Leibler divergence for stochastic processes. The number of the measurement points is a random variable. This new KL divergence is well defined for stochastic processes which are generated from different network architectures. Based on the new KL divergence, the authors developed a scalable variational inference using the VAE to generate approximate posterior stochastic processes.

**Limitations And Societal Impact:**

Yes, they are shown in the checklist.

**Main Review:**

Overall, I think the paper could be accepted, but requires a revision for the algorithm part in Section 3.2 to 3.4.

# Pros
-  This paper provides a new KL divergence, which is valid for different network architectures.
-  Numerical experiments, especially for the structured implicit priors show strong evidence that the proposed method is useful numerically.

# Cons
-  The theoretical part of the paper is hard to follow for a non-expert. Especially, in the proof of Proposition 4, I could not follow the discussion of the sequence of VAEs.
-  Algorithmic part is hard to follow at first since each detailed technique is presented one by one.

# Comments and Questions:
- How strong are assumptions in Proposition 2 ? In the proposition, the compactness of the supports of $p(\theta)$ and $q(\Gamma)$  are assumed. I thought these p and q are Gaussian distributions or Inverse Gamma distributions in standard Bayesian neural networks. Then, their supports are not compact, thus it violates the assumption.
- Is there any other example for the distribution of $n$ other than a geometry distribution ? And is there any reason that using a geometry distribution is important numerically ?
- In numerical experiments, how did authors construct the compact space $\mathcal{T}$ for the input space from the data set ?
- Why should we distill the prior distribution $p(f)$ ?
- Is it possible to compare the KL divergence defined in (2) and (3) numerically ?
- In proposition 4, using VAE is justified regarding MMD distance. On the other hand, in variational inference, we are working on KL divergence, not MMD. Is it possible to justify using VAE based on KL divergence under some mild conditions?
- I think experiments for the structured implicit priors in Sec 5.1 strongly support the usefulness of the proposed method. I recommend using the real dataset for this task to support your proposed method.




**Time Spent Reviewing:**

3 to 4 hours

---

> ### Author Response · Authors · 2021-08-10
> **Response to Reviewer 4izH**
>
> Thank you very much for your positive opinions for our work. We are pleased to see that you appreciate the technical novelty and the practical impact of our work. In the following, we will try to address each question in the review.
>
> > **Q1: The theoretical part of the paper is hard to follow for a non-expert. Especially, in the proof of Proposition 4, I could not follow the discussion of the sequence of VAEs.**
>
> **A1:** Thanks for the comment, we will try to provide more background information in the appendix, as well as making the discussion more clear.
>
> > **Q2: Algorithmic part is hard to follow at first since each detailed technique is presented one by one.**
>
> **A2:** Thank you for your input. In revision, we will improve the presentation, and add illustrations of the algorithm to make it more clear to more general audiences.
>
> > **Q3: How strong are assumptions in Proposition 2 ?**
>
> **A3:** Assumptions in proposition 2 are merely sufficient conditions. The conclusion of Proposition 2 still holds as long as the infinite series $\sum_{n=1}^\infty p(n) \mathbb{E}_{\mathbf{X}_n \sim U(\mathcal{T}^n)} KL[q(\mathbf{f}^{\mathbf{X}_n})|| p(\mathbf{f}^{\mathbf{X}_n})] $ is convergent. For example, using a BNN with Gaussian weights as prior does not break the proof in B.2. We will add discussions on BNNs in revision. Also, as shown in B.3, even if the finite-dimensional latent representation assumption in Proposition 2 is violated (for example one of the distribution is GPs), similar results still hold. In revision, we will provide more discussion regarding the practical implications/extensions of proposition 2 in the appendix.
>
> > **Q4: Is there any other example for the distribution of other than a geometry distribution? And is there any reason that using a geometry distribution is important numerically?**
>
> **A4:** The idea behind Proposition 2 is that, we need to choose $p(n)$ such that $\sum_{n=1}^\infty p(n) \mathbb{E}\_{\mathbf{X}\_n \sim U(\mathcal{T}^n)} KL[q(\mathbf{f}^{\mathbf{X}\_n})|| p(\mathbf{f}^{\mathbf{X}\_n})] $ is convergent as a infinite series. For example, in Appendix B.2 (case 2) we have shown that in order for $\sum_{n=1}^\infty p(n) \mathbb{E}\_{\mathbf{X}\_n \sim U(\mathcal{T}^n)} KL[q(\mathbf{f}^{\mathbf{X}\_n})|| p(\mathbf{f}^{\mathbf{X}\_n})] $ to be absolute convergent, $p(n)$ only need to satisfy $\lim_{n \rightarrow \infty}p(n+1)/p(n) < 1 $. We merely choose geometry distribution for convenience.
>
> > **Q5: In numerical experiments, how did authors construct the compact space $\mathcal{T}$ for the input space from the data set?**
>
> **A5:** It depends on specific tasks. If we know the range of the input $x$, we can directly set $\mathcal{T}$ to be such interval. For example, in synthetic datasets of Experiment 5.1 (Figure 2), we already know that the input $x$ lies in the interval between 0 and 1, therefore $\mathcal{T} = [0,1]$. If we don' t know the range of the inputs, then we can simply set $\mathcal{T} = [x_{min},x_{max}]$, where $x_{min}$ and $x_{max}$ are the sample min and max of the input dataset. We will clarify the choice of  $\mathcal{T}$ in Appendix.
>
> > **Q6: Why should we distill the prior distribution $p(f)$ ?**
>
> **A6:** This is because in many cases, $p(f)$ is intractable to evaluate (such as the structured implicit priors considered in Experiment 5.1). Therefore, we need a way to distill the information of the intractable prior $p(f)$ into a tractable surrogate, such that we can evaluate and optimize the grid-functional ELBO $\mathcal{L}^{grid}$ tractable. In the end, we will expect that the posterior distribution produced by our method is able to approximate the true posterior well (Figure 2).
>
> > **Q7: Is it possible to compare the KL divergence defined in (2) and (3) numerically?**
>
> **A7:** Firstly, (2) and (3) are essentially different divergence measures, therefore it does not make much sense to directly compare their scales numerically. Secondly, as pointed out by (Burt et al., 2020), the divergence (2) might take infinite values for the settings considered in Proposition 2. Furthermore, even if (2) is finite, it still requires a maximization procedure (supremum over all possible $n$ and $\mathbf{X}_n$), which is often not tractable.
>
> > **Q8: In proposition 4, using VAE is justified regarding MMD distance.  Is it possible to justify using VAE based on KL divergence under some mild conditions?**
>
> **A8:** This is a good question. Indeed, we justified our choice of parameterizations using MMD distance. This is because, in our proof in proposition 4, we need to utilize the "metrization of weak convergence of probability measure'' property of MMD distances, which is described in B.4. Unfortunately, this does not hold for KL divergence. Therefore, currently, we are not able to comment on whether it is possible to prove similar results with KL divergences. However, it is definitely an interesting topic to look at in the future.
>
> > **Q9: I think experiments for the structured implicit priors in Sec 5.1 strongly support the usefulness of the proposed method. I recommend using the real dataset for this task to support your proposed method.**
>
> **A9:** Thanks for the comment. We will try to add real datasets that satisfy piecewise constant/linear properties in revision.
>
> **References**
>
> Burt, D. R., Ober, S. W., Garriga-Alonso, A., \& van der Wilk, M. (2020). Understanding variational inference in function-space. arXiv preprint arXiv:2011.09421.

---

### Official Review · Reviewer_w44o · 2021-07-16

**Rating:** 6
**Confidence:** 5

**Summary:**

This paper proposes a new framework for conducting approximate Bayesian inference in the context of function-space optimization. The method described is fairly general and with good theoretical justifications to some of the most important claims here. To achieve their goal, the authors define a new approximate objective function with an alternative divergence to the regular KL  and propose the SPGs as a generalization over previous work. Even though there is some important information missing in the experimentation phase, the performance of the method seems overall much better than previous state-of-the-art models in many contexts. This needs to be addressed since it constitutes an important issue, but the method is promising and interesting.




**Limitations And Societal Impact:**

The societal impact of research such as this one is mostly restricted to the community of approximate inference, especially to those interested in function space optimization. Therefore, I deem there is no potential negative societal impact.

**Main Review:**

In this paper, the authors propose a new approach to inference in functional space, which has been an important issue of research for the past few years. This is done by introducing an alternative objective function with different properties to the ones of the regular functional-ELBO. The change in the objective function is effectively a newly proposed divergence (grid-functional KL divergence), which could be interpreted as the expected value of the KL divergence between functions sampled from two stochastic processes, evaluated at a random number of measurement locations. The key difference here is that the number of sampling points is a random variable instead of remaining fixed, proposing a grid approximation to the KL divergence. Combining this with the SPG, the authors extend on the formulation by previous work avoiding some of the restrictions of previous methods.

The results presented in the paper show that the method behaves well and performs better overall than the rest of the methods compared it is being against in the experiments conducted. It represents a fairly general approach (although with certain important similarities to previous literature work) that could help further the comprehension and usage of function space optimization methods.  However, although the results are promising, there are some important experimental concerns that should be addressed in order to have a clearer picture of the capabilities of the proposed framework.

# Pros

* There is considerable theoretical work being conducted to justify a lot of the properties mentioned of this new approach. Although a bit tough at times, the descriptions in the main text as well as in the appendix help a lot with the comprehension of the proposal.

* The framework constructed is general enough to be used with several methods. This flexibility could be interesting to extend the formulation with the usage of new methods.

* The performance of the method seems to improve on the previous state-of-the-art, both when compared with some of the best methods in weight and function space optimization.

* The paper, as well as the appendix material, is well written and easily comprehensible for the most part. Although some sections need a careful read, in most cases it is written in a very clear manner.

# Cons

* If L^grid >= L^functional, then optimizing it does not mean we are also optimizing L^functional since it makes a weaker lower bound definition (since it is higher, maximizing it would not necessarily imply maximizing L^grid). Although the method seems to perform well, this point needs to be addressed more carefully due to its crucial importance in the development of the paper and the training of the model.

* The experiments are conducted using the same prior, which is good as a reference. Since this approach also supports other priors, it would be interesting to see how other methods would perform in comparison given their respectively "best" prior (as described in their respective papers). For example, for f-BNN, the usage of a piece-wise implicit prior may not be optimal since that method is not able to train such type of priors, and thus a GP (or a sparse GP) could help improve its performance w.r.t. what is seen in the experiments.

* Regarding the regression experiments, there is important information not being reported in the main text nor in the appendix. It would be very helpful to have a table with the RMSE (or some other alternative error metric) for completeness, while also having another alternative measure other than the log-likelihood to address the goodness-of-fit of the results (see [1]).

* Through the paper there are some claims about the scalability of the method. However, the material provided lacks some kind of information on the convergence rate and time of this method in comparison to the other ones. To better support these claims, please provide a convergence study for the methods here.




# Notes

* Does this formulation generalize over implicit stochastic processes (as described in the paper of Variational Implicit Processes), or is it a more restricted description form for the stochastic processes involved? Specifically, clarify this w.r.t. the description of the functions sampled from both stochastic processes in Proposition 2 and the implications made in the following paragraph (those very same properties could be assigned to implicit processes as well).

* Also regarding the experiments, although an extensive comparison with other methods is conducted, in the case of weight-space optimization there are some methods that currently perform better than the ones present here, such as the one described in [2]. For a more comprehensive description of the performance, it would be convenient to include these approaches.

* How flexible could be the resulting predictive distributions that the method is able to produce? Could you try to provide any tests to showcase this? As far as the formulation goes, the predictive distributions could be pretty general, but it would help if there was any experimental confirmation of this fact.

* Are there any concerns with the problems that may arise due to using a sampling-based approach for the estimation of the KL? For f-BNN it could be seen that they would be somewhat affected by the dimensionality of the data precisely due to this fact.

* Please provide the information on the computational resources employed to conduct the experiments.

*Minor details:*

* Although it is comprehensible as is, maybe the notation in the equations of Proposition 2 could be a little more different so that there is a clearer distinction between the functions sampled from $q$ and $p$

* Double ":" in line 85

* The enumeration in the second paragraph of the introduction could appear differently so it helps reading the text, either by enclosing properly the (i) and (ii) or by separating each point with a different item in an enumerated list.



##  References

[1] Gneiting, T., & Raftery, A. E. (2007). Strictly proper scoring rules, prediction, and estimation. Journal of the American Statistical Association, 102(477), 359-378.

[2] Santana, S. R., & Hernández-Lobato, D. (2020). Adversarial α-divergence minimization for Bayesian approximate inference. Neurocomputing.




**Time Spent Reviewing:**

4

---

> ### Author Response · Authors · 2021-08-10
> **Response to Reviewer w44o**
>
> Thank you for your encouraging review and valuable suggestions to improve.  We are particularly pleased that you appreciate the significance of our work and the clarity of the presentation. In the following, we will try to address each question in the review.
>
> > **Q1: If $\mathcal{L}^{grid} >= \mathcal{L}^{functional}$, then optimizing it does not mean we are also optimizing $\mathcal{L}^{functional}$ since it makes a weaker lower bound definition (since it is higher, maximizing it would not necessarily imply maximizing $\mathcal{L}^{functional}$).**
>
> **A1:**  It is true that  "optimizing $\mathcal{L}^{grid}$ does not mean we are also optimizing $\mathcal{L}^{functional}$''. However, we would like to emphasize that $\mathcal{L}^{functional}$ is *not* the ultimate objective that we are trying to improve. Our paper proposed to use $\mathcal{L}^{grid}$ as a new objective to *replace* $\mathcal{L}^{functional}$, instead of *approximating* $\mathcal{L}^{functional}$. As discussed both in our paper and in Burt et al., 2020, $\mathcal{L}^{functional}$ (Sun et al., 2018) could be ill-defined for many interesting cases (i.e., $\mathcal{L}^{functional}$ be comes $-\infty$).  Therefore, we propose to optimize $\mathcal{L}^{grid}$, which is 1) more well-behaved than $\mathcal{L}^{functional}$ (Proposition 2); and 2), it is a valid variational objective function (Proposition 1 and 3). Given those results, $\mathcal{L}^{functional}$ becomes less relevant in our paper.
>
> > **Q2:  Since this approach also supports other priors, it would be interesting to see how other methods would perform in comparison given their respectively "best" prior (as described in their respective papers).**
>
> **A2:** Thanks for the suggestion. If we understood this comment correctly, the reviewer seems to suggest that we may consider performing some kind of "optimization'' w.r.t to the prior hyperparameters (e.g., parameters of the BNN prior weight distributions) or even the types of the priors (e.g., piece-wise implicit or GP). While this is a very interesting suggestion, our paper mainly treats FVI/f-BNN/VIPs/MFVIs, etc as *inference methods* rather than *standalone models*. From this perspective, we are more interested in the case where the prior is fixed (hence the ground truth posterior is given).
>
> That being said, if we were to treat (the predictive distribution of) each inference method as a standalone model, then indeed it makes sense to seek the optimal prior for each method. We will try to include a discussion on the second perspective in revision.
>
> >**Q3: It would be very helpful to have a table with the RMSE (or some other alternative error metric) for completeness, while also having another alternative measure other than the log-likelihood to address the goodness-of-fit of the results (see [1]).**
>
> **A3:** Thanks for the reference. We will add more metrics for regression tasks to the appendix in revision.
>
> > **Q4: Through the paper, there are some claims about the scalability of the method. However, the material provided lacks some kind of information on the convergence rate and time of this method in comparison to the other ones.**
>
> **A4:** Thanks for the comment. We will add this information to the appendix in revision. We originally did not provide the convergence time analysis, as it seems to be not very common in BNN/VI literature. As a preliminary result, in Appendix D.4 and D.3, we have included the computational time on contextual bandits and synthetic datasets, comparison among different methods.
>
> > **Q5: Regarding the experiments, although an extensive comparison with other methods is conducted, in the case of weight-space optimization there are some methods that currently perform better than the ones present here, such as the one described in [2]. For a more comprehensive description of the performance, it would be convenient to include these approaches.**
>
> **A5:** Thanks for pointing out the reference. We will cite the AADM paper and include the results of AADM in revision. Note that our focus in experiments is not to include every state-of-the-art weight-space methods, but to consider those that are the most commonly used.
>
>
>
> > **Q6: Does this formulation generalize over implicit stochastic processes, or is it a more restricted description form for the stochastic processes involved? Specifically, clarify this w.r.t. the description of the functions sampled from both stochastic processes in Proposition 2 and the implications made in the following paragraph.**
>
>
> **A6:** The processes $p$ and $q$ considered in Proposition 2 are in fact the same as the definition of implicit process in Ma et al., 2019, under the finite-dimensional $\mathbf{z}$ regime. This can be shown by noticing the fact that $\Theta$ is just $\mathbf{z}$ in (Ma et al., 2019), and $p(g|\mathbf{x},\Theta)$ can be deterministic (see Case 2 in Appendix B.2). The major difference here is that we mainly consider the implicit processes that have finite-dimensional latent randomness, such that the grid-functional KL between them is finite. On the contrary, in (Ma et al., 2019), their definition does not have such assumption since they did not consider any "proper'' functional KL divergences at all (they only considered KL divergences on a fixed grid).
>
>
> > **Q7: How flexible could be the resulting predictive distributions that the method is able to produce? Could you try to provide any tests to showcase this?**
>
> **A7:** Great question. In fact, when designing our experiments, we are always keeping the idea of `"testing the flexibility and quality of predictive distributions'' in mind.
>
> * Intuitively, in order to examine the quality/flexibility of predictive distribution $q(f)$ produced by a inference method (which is also the approximate posterior distribution $q(f)$), the most straightforward method is to: 1), specify a complex high dimensional stochastic process prior, $p(f)$; 2), specify a (large) number of observations, $\mathcal{D}$; 3), use FVI (or whatever inference method of choice) to generate the predictive distribution $q(f|\mathcal{D})$, by minimizing $\mathrm{D}_{KL}^{grid}[q||p(f|\mathcal{D})]$ (or maximizing $\mathcal{L}^{grid}$); 4), compute the true posterior/predictive distribution $p(f|\mathcal{D})$; and finally 5), compare how well $q(f|\mathcal{D})$ is able to approximate $p(f|\mathcal{D})$.
>
> * To compare how well $q(f|\mathcal{D})$ is able to approximate $p(f|\mathcal{D})$, we consider the following two scenarios: **scenario i)**, low dimensional complicated prior + small data $\mathcal{D}$ regime, where $p(f|\mathcal{D})$ is tractable via MCMC methods; and **scenario ii)**, high dimensional complicated prior + large data $\mathcal{D}$ regime, where $p(f|\mathcal{D})$ is not tractable.
>
> * For **scenario i)**, since sampling from $p(f|\mathcal{D})$ is computationally managable, we can directly compare $q(f|\mathcal{D})$ to $p(f|\mathcal{D})$. This is exactly the testing procedure used in Experiment 5.1 (Figure 2), which indeed showcased the flexibility and approximation quality of the predictive distribution of FVI.
>
> * For **scenario ii)**, $p(f|\mathcal{D})$ is often intractable. we could split $\mathcal{D}$ into $\mathcal{D}\_{train} = \\{\mathbf{y}\_{train}, \mathbf{X}\_{train} \\}$ and $\mathcal{D}\_{test} = \\{ \mathbf{y}\_{test}, \mathbf{X}\_{test} \\}$, and calculate the predictive accuracy metrics on $\mathcal{D}\_{test}$ using $q(\mathbf{y}\_{test}|\mathcal{D}\_{train}, \mathbf{X}\_{test}) = \int\_{f} p(\mathbf{y}\_{test}|f, \mathbf{X}\_{test}) q(f|\mathcal{D}\_{train}) df  $, which could be a proxy for the approximation accuracy. This is exactly what we did in Experiment 5.2 and 5.4, where we used complicated BNN priors + high dimensional large scale datasets, and compared accuracies and llhs on  $q(\mathbf{y}\_{test}|\mathcal{D}\_{train}, \mathbf{X}\_{test})$. This shows that the flexibility of $q(f|\mathcal{D}\_{train})$ produced by FVI is at least on par with other standard methods.
>
> > **Q8: Are there any concerns with the problems that may arise due to using a sampling-based approach for the estimation of the KL? For f-BNN it could be seen that they would be somewhat affected by the dimensionality of the data precisely due to this fact.**
>
> **A8:** Compared with f-BNNs, we believe that our method is less affected by the dimensionality of the data, since we utilized a latent representation to estimate the grid-functional KL (proposition 5 and 6). However, we do agree that sampling-based estimation could be a limiting factor, and this is, in fact, one of the focuses in our future/ongoing research.
>
> > **Q9: Please provide the information on the computational resources employed to conduct the experiments.**
>
> **A9**: Synthetic experiment/contextual bandits are mostly run on a laptop using only the CPUs. Other experiments are mostly done on a machine with NVIDIA Quadro P5000 GPU plus 30GB of memory. We will add this info in revision.
>
> **References**
>
>  Ma, C., Li, Y., \& Hernández-Lobato, J. M. (2019, May). Variational implicit processes. In International Conference on Machine Learning (pp. 4222-4233). PMLR.

---

### Decision · Program_Chairs · 2021-09-27

**Decision:**

Accept (Poster)

**Comment:**

This paper proposes a method for approximate Bayesian inference in function space using a novel form of the KL divergence between stochastic processes, the grid-functional KL divergence, where the number of measurement points is a random variable. This divergence has better properties than those used in previous work. All the reviewers recommend acceptance and I agree.